# Domain gain or loss in a fungal chitinase enables specialization towards antagonism or immune suppression

Ruben Eichfeld[1,2], Asmamaw B. Endeshaw[1], Margareta J. Hellmann [3], Taim Nassr [1], Bruno M. Moerschbacher [3] & Alga Zuccaro [1,2] ✉

Effector proteins orchestrate interactions that determine host compatibility and microbial competition. Among these are microbial chitinases, widespread enzymes involved in nutrient acquisition, fungal cell-wall remodelling, and antagonism, yet how their functions diversify across ecological roles remains unclear. Here, we show that modular domain variation within a conserved GH18 chitinase family enables specialization between microbial antagonism and host immune suppression in the beneficial root endophyte *Serendipita indica*. The chitinase *Si*CHIT, bearing a C-terminal CBM5 domain, is induced during fungal confrontation and inhibits growth of the phytopathogen *Bipolaris sorokiniana*, thereby protecting host roots. Deleting CBM5 abolishes this antagonistic activity, whereas grafting CBM5 onto its paralog *Si*CHIT2, which lacks this domain, confers antifungal function. In contrast, *Si*CHIT2 is induced during root colonization and suppresses chitin-triggered immunity, promoting host compatibility. These results show how domain loss and regulatory divergence can reprogram antimicrobial enzymes into immune-suppressive effectors in beneficial plant-associated fungi.

Microbial effector proteins shape the outcome of host–microbe and microbe–microbe interactions across diverse biological systems. A central question in molecular ecology and symbiosis research is how new ecological functions emerge from existing protein architectures[1]. Structural modularity enables proteins to acquire new functions by altering domain composition through duplication, gain and loss, or recombination[2,3]. In microbial effectors, such architectural variations potentially drive shifts in ecological roles, including inter-microbial antagonism or host immune suppression. However, direct mechanistic evidence linking domain changes to functional repurposing remains limited, particularly in mutualistic fungi.

Mutualistic fungi are ecologically important members of the rhizosphere microbiome, where they support plant health and shape microbial community structure and function[4]. This diverse group includes ectomycorrhizal and arbuscular mycorrhizal fungi, as well as beneficial root endophytes. Unlike mycorrhizal fungi, root endophytes often colonize plant roots without forming specialized symbiotic structures and, depending on the host and environmental context, can enhance plant performance by promoting nutrient uptake, water acquisition, tolerance to abiotic stress, and suppression of microbial pathogens that threaten the root niche[5–10].

These fungi exemplify the evolutionary plasticity of fungal lifestyles, flexibly occupying ecological niches along the saprotrophy-to-symbiosis continuum. To colonize plant roots and maintain their niche, beneficial root endophytes must overcome two key challenges: evasion of host immunity and competition with microbial antagonists and plant pathogens[11,12]. This dual challenge is increasingly understood as part of a broader ecological strategy, supported in some cases by synergistic interactions with bacterial members of the core microbiota[8,13]. Both immune evasion and microbial competition are mediated by secreted effector proteins[11,14–17]. Effector-mediated immune evasion often involves the degradation or masking of

[1]University of Cologne, Institute for Plant Sciences, Cologne, Germany. [2]Cluster of Excellence on Plant Sciences (CEPLAS), Cologne, Germany. [3]University of Münster, Institute of Biology and Biotechnology of Plants, Münster, Germany. ✉e-mail: azuccaro@uni-koeln.de

microbe-associated molecular patterns (MAMPs), such as β-1,3/1,6-glucans and chitin, to prevent recognition by host pattern recognition receptors[18–23]. In parallel, effectors can modulate the surrounding microbiota by suppressing microbial competitors, a strategy well documented in plant pathogens[11,14,16] and potentially present in mutualistic fungi as a mechanism for niche defense.

The beneficial root endophyte *Serendipita indica* (*Si*), a member of the widespread fungal order Sebacinales (Basidiomycota), colonizes a broad range of host plants and is known for its ability to promote growth and suppress plant immunity while protecting the host from pathogens[8,24–29]. Colonization is confined to the rhizoplane, epidermis, and cortex, and depends on an intact host immune system[30]. In addition to its host-associated functions, *Si* exhibits strong antifungal activity in the rhizosphere, particularly against the pathogen *Bipolaris sorokiniana* (*Bs*), the causal agent of spot blotch and root rot in cereals[8,25,31]. This antagonism involves multiple molecular factors, with a GH18-CBM5 endochitinase (*Si*CHIT) playing a prominent role by restricting *Bs* growth and recapitulating the protective effect of *Si in planta*[31].

GH18 chitinases, found across all fungal phyla, cleave β-1,4-glycosidic bonds in chitin, a major structural component of fungal cell walls. These enzymes fulfill diverse roles in nutrient scavenging, hyphal remodeling, and fungal antagonism[32,33]. Their catalytic activity depends on a conserved DxxDxDxE motif, which defines the substrate-assisted catalytic mechanism characteristic of this family[34,35]. While some chitinase-derived effectors have lost enzymatic activity and evolved to sequester MAMPs[36], others retain catalytic function and are functionally enhanced by accessory domains such as carbohydrate-binding modules (CBMs)[37–39]. These non-catalytic domains increase substrate affinity and specificity, particularly for crystalline or polymeric forms of chitin[40,41]. *Si*CHIT features a C-terminal CBM5 domain, a modular architecture found predominantly in bacteria and Basidiomycota but absent in Ascomycota, suggesting a lineage-specific evolutionary adaptation[31,33].

Here, we investigate the functional significance of CBM5 in *Si*CHIT and examine how domain architecture modulates effector activity, enabling a switch between microbial antagonism and immune evasion. We show that the CBM5 domain is essential for the antifungal and plant-protective function of *Si*CHIT. This enzyme is specifically expressed during fungal competition and restricts the growth of the phytopathogen *Bs* in the rhizosphere, thereby protecting host roots. In contrast, its closely related paralog *Si*CHIT2 lacks CBM5 and is induced during root colonization, where it suppresses chitin-triggered immune responses by preventing host-derived reactive oxygen species accumulation. Through domain-swapping experiments, we demonstrate that CBM5 functions as a modular determinant capable of switching a chitinase's ecological role from microbial antagonism to host immune suppression, while retaining its core chitinolytic activity. Together, our findings support a model in which effector specialization in mutualistic fungi may arise through gene duplication, domain loss, and transcriptional divergence, enabling the functional repurposing of an antimicrobial enzyme for host adaptation.

## Results

### CBM5 enhances binding to crystalline chitin and fungal cell wall without altering product specificity

To investigate the role of the C-terminal CBM5 domain in *Si*CHIT (PIIN_03543; Pirin1_74346), we generated three recombinant variants, all lacking their native signal peptides: full-length *Si*CHIT, a catalytically inactive mutant (*Si*CHIT^E196Q), and a truncated version lacking the CBM5 domain (*Si*CHIT^−CBM5), which retained the proline-rich linker. The truncated construct was produced by excluding the final 141 bp from the coding sequence. All recombinant proteins were purified via affinity chromatography and verified by SDS−PAGE (Supplementary Fig. 1).

We first assessed chitinase activity on insoluble crab shell chitin by quantifying reducing sugars after 20 h of incubation (Fig. 1A). This assay revealed no major differences between full-length *Si*CHIT and *Si*CHIT^−CBM5. However, time-course experiments showed that the presence of CBM5 enhanced early hydrolysis on crab shell chitin (Fig. 1B), suggesting that the domain increases the reaction rate during initial substrate hydrolysis. A similar trend was observed for chitinolytic activity on chitin azure (CA) (Fig. 1C, D) but not for activity on >75% de-acetylated chitosan, where *Si*CHIT and *Si*CHIT^−CBM5 displayed similar reaction rates already at the onset of the reaction (Fig. 1E).

To evaluate whether CBM5 influences product outcome, we analyzed the hydrolysates of crab shell chitin by mass spectrometry. Both *Si*CHIT and *Si*CHIT^−CBM5 produced similar oligomer profiles dominated by fully or partially acetylated dimers (A2 or A1D1), indicating that CBM5 does not alter product specificity (Fig. 1F, Supplementary Fig. 2). Notably, substantial amounts of other partially deacetylated products, such as A1D2 and A2D1, were also detected. This likely reflects the fact that crab shell chitin is not fully acetylated; deacetylated regions may form stretches with higher solubility and increased accessibility, leading to their preferential cleavage by the enzymes. Our mass spectrometric analysis revealed a residual activity of *Si*CHIT^E196Q that was not detectable with the other, less sensitive activity assays using crab shell chitin or CA. We also analyzed product profiles on β-chitin (Supplementary Fig. 3). While truncation of the CBM5 domain did not affect the product profile on this substrate, slight differences were observed depending on the substrate. These differences can be attributed to the less rigid structures of β-chitin, which result from its more loosely packed polymer folding.

To assess whether CBM5 affects subsite preferences, we analyzed the hydrolysis products of chitosans with defined degrees of acetylation (DA). *Si*CHIT^−CBM5 displayed the same requirement for an acetylated unit at the −1 subsite and no detectable shifts at adjacent positions compared to *Si*CHIT[31], confirming that CBM5 does not influence catalytic specificity (Fig. 1G, Supplementary Fig. 4).

To determine the role of CBM5 in substrate interaction, we performed pull-down assays using crab shell chitin, >75% de-acetylated chitosan, and lyophilized, washed *Bs* mycelium. *Si*CHIT^−CBM5 showed reduced binding to crab shell chitin compared to full-length *Si*CHIT, whereas no difference was observed with chitosan, suggesting that CBM5 enhances binding specifically to acetylated, crystalline substrates (Fig. 1H; Supplementary Fig. 5). This is consistent with our observation that CBM5 increases the initial degradation rate on crab shell chitin but not on chitosan (Fig. 1B, E). Binding to *Bs* mycelium was likewise reduced in the absence of CBM5, indicating that the domain contributes to effective fungal cell wall targeting (Fig. 1H).

Together, these results show that although CBM5 is dispensable for full catalytic activity and does not alter product specificity, it enhances early hydrolysis rates on chitin and increases substrate binding to crystalline chitin and fungal mycelium. This suggests that CBM5 improves functional efficiency under biologically relevant conditions, likely contributing to the role of *Si*CHIT in microbial competition.

### CBM5 is required for antifungal activity and plant protection

We previously demonstrated that *Si*CHIT restricts the growth of *Bs* and reduces its colonization of plant roots through its chitinolytic activity[31]. However, the specific contribution of the CBM5 domain to this antifungal function remained unresolved. To test whether CBM5 is required for *Bs* inhibition, we incubated *Bs* spores with the full-length *Si*CHIT, the catalytically inactive mutant *Si*CHIT^E196Q, the truncated variant lacking CBM5 *Si*CHIT^−CBM5, or the empty vector control Ev. As expected, *Bs* spore germination was significantly reduced in the presence of active *Si*CHIT, but not with *Si*CHIT^E196Q. *Si*CHIT^−CBM5 showed a similar lack of inhibitory effect as *Si*CHIT^E196Q and had only a minimal effect compared to the Ev control (Fig. 2A).

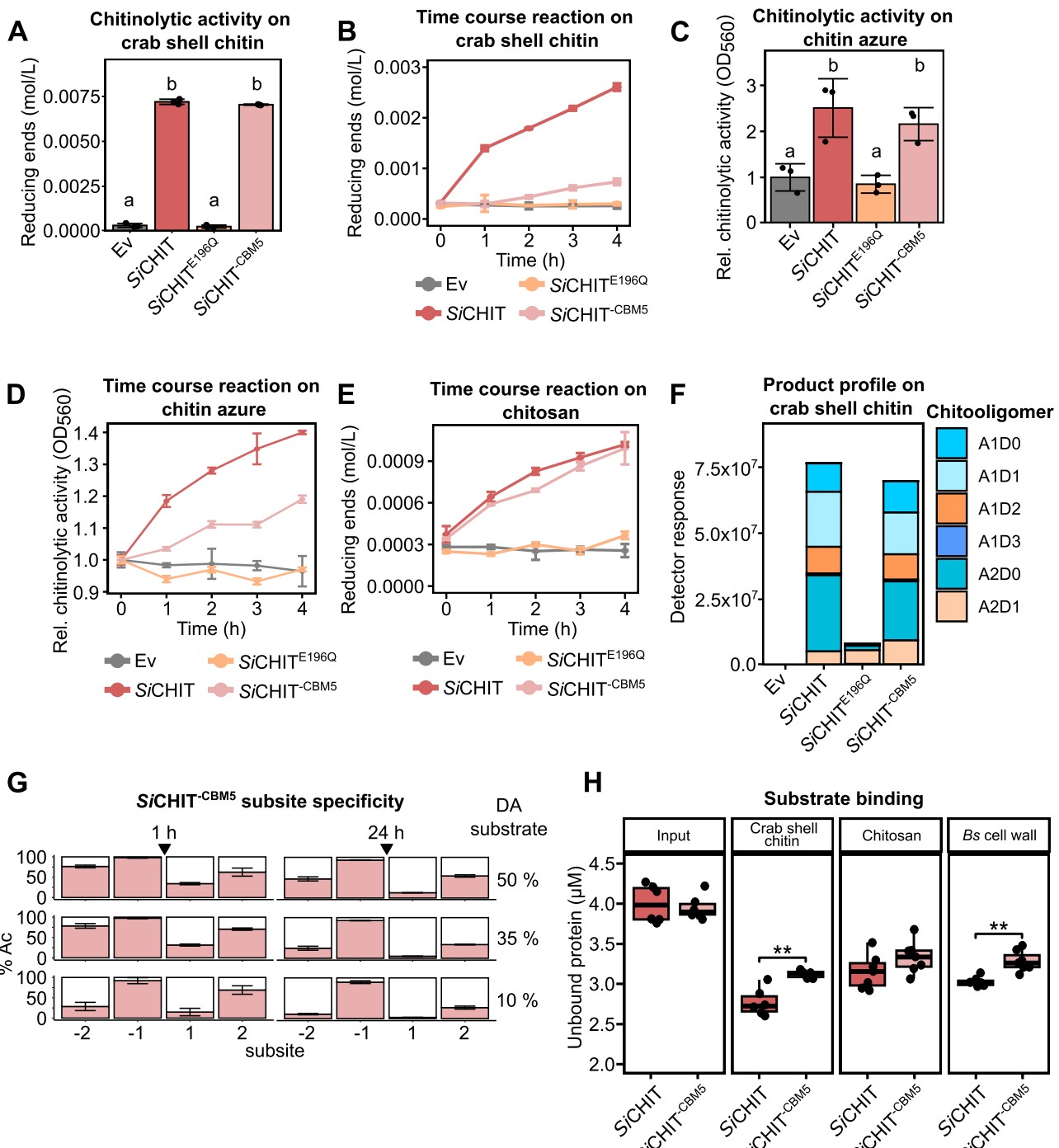

**Fig. 1 | CBM5 enhances binding to fungal cell wall and crystalline chitin and promotes hydrolysis without affecting product specificity. A** Chitinase activity of recombinant chitinases on crab shell chitin at 10 μM and after 20 h at 28 °C. The relative amount of reducing ends was measured at 540 nm using DNSA (mean ± SD, $n = 3$). **B** Chitinase activity of recombinant chitinases on crab shell chitin at 10 μM measured over 4 h at 28 °C. The relative amount of reducing ends was measured at 540 nm using a DNSA assay (mean ± SD, $n = 4$). **C** Chitinase activity of recombinant chitinases at 10 μM after 20 h at 28 °C on chitin azure (CA). Supernatants were quantified at 560 nm and normalized to Ev (mean ± SD, $n = 3$). **D** Chitinase activity of recombinant chitinases on CA at 10 μM measured over 4 h at 28 °C and normalized to $t_0$ (mean ± SEM, $n = 3$). **E** Chitinase activity of recombinant chitinases on >75% de-acetylated chitosan at 10 μM measured over 4 h at 28 °C. The relative amount of reducing ends was measured at 540 nm (mean ± SD, $n = 3$). **F** Product profile of recombinant chitinases on crab shell chitin. Hydrolysis products were identified and quantified via MS. A = acetylated, D = de-acetylated unit ($n = 1$).

**G** Subsite preference of $Si$CHIT$^{-CBM5}$ measured by MS. Chitosan of three degrees of acetylation (DA) was hydrolyzed for 1 or 24 h. The frequency of acetylated units at the −2 to +2 subsites of $Si$CHIT$^{-CBM5}$ was determined. The black arrow indicates the gylcosidic bond between the −1 and +1 subsite that is cleaved by the enzyme (mean ± SD, $n = 3$). **H** Substrate-binding ability of $Si$CHIT and $Si$CHIT$^{-CBM5}$ on crab shell chitin, >75% de-acetylated chitosan, or lyophilized $Bs$ mycelium. Binding was assessed by measuring the protein amount in supernatants compared to input ($n = 6$ for input and crab shell chitin; $n = 7$ for chitosan and $Bs$ cell wall). Limits of the boxplots represent the 25th–75th percentile, the horizontal line represents the median, and the whiskers represent the minimum/maximum values without outliers. Statistical analysis: All displayed data ($n$) is derived from independent biological replicates. A and C Different letters indicate significant differences according to one-way ANOVA followed by Tukey's HSD test (adjusted $p$-value < 0.05). **H** Two-sided Student's $t$-test ($P$-value: * <0.05; ** <0.01).

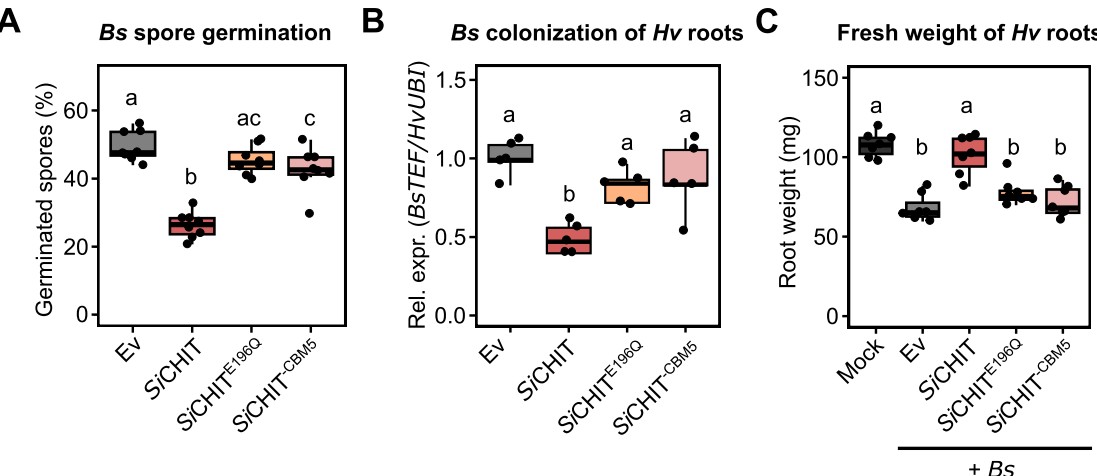

**Fig. 2 | The CBM5 domain is essential for the antifungal and plant-protective activity of *Si*CHIT. A** Relative *Bs* spore germination six h post incubation in sterile chamber slides with the Ev control or 10 µM of *Si*CHIT, *Si*CHIT$^{E196Q}$ or *Si*CHIT$^{-CBM5}$ ($n = 8$). **B** Colonization of *Hv* roots by *Bs* at three dpi inferred from relative expression of the fungal housekeeping gene *BsTEF* compared to the *Hv* housekeeping gene *HvUBI* inferred by qPCR using the $2^{-\Delta CT}$ method. Colonization values were normalized to the Ev control. *Bs* spores were either pre-treated with the Ev control or 10 µM of recombinant chitinases for 20 h at 28 °C (mean ± SD, $n = 5$). **C** *Hv* root fresh weight at three dpi with *Bs* spores. *Bs* spores were either pre-treated with the Ev control or 10 µM of recombinant chitinases for 20 h at 28 °C ($n = 7$). Limits of the boxplots represent the 25th–75th percentile, the horizontal line represents the median, and the whiskers represent the minimum/maximum values without outliers. Statistical analysis: All displayed data ($n$) is derived from independent biological replicates. Different letters indicate significant differences according to one-way ANOVA followed by Tukey's HSD test (adjusted *p*-value < 0.05).

We next examined whether the loss of antifungal activity translates into reduced protection of host roots. *Bs* spores treated with the same chitinase variants were used to inoculate the roots of the *Bs* host *Hordeum vulgare* (*Hv*). As previously shown, *Si*CHIT-treated spores led to a significant reduction in *Bs* colonization, as measured by fungal transcript levels in roots. In contrast, *Si*CHIT$^{E196Q}$ and *Si*CHIT$^{-CBM5}$ had no effect (Fig. 2B). Correspondingly, only *Si*CHIT mitigated the reduction in root biomass caused by *Bs* infection (Fig. 2C). To test whether this effect is conserved in other hosts, we repeated the experiment using *Arabidopsis thaliana* (*At*). *Si*CHIT reduced *Bs* biomass and preserved root elongation, while *Si*CHIT$^{E196Q}$ did not. *Si*CHIT$^{-CBM5}$ had a weak protective effect, albeit significantly lower than the full-length enzyme (Supplementary Fig. 6).

Together, these results demonstrate that both chitinase activity and the CBM5 domain are required for *Si*CHIT's full antifungal and plant-protective functions. The loss of protection observed in the CBM5-lacking variant indicates that this domain enhances fungal cell wall engagement and is critical for effective niche defense.

### A CBM5-lacking paralog, *Si*CHIT2, is transcriptionally specialized for host colonization

GH18-CBM5 chitinases are absent in Ascomycota but occur frequently in the Agaricomycetes (Basidiomycota), where gene copy number varies across species with different ecological lifestyles[31]. To better define the structural requirements for antifungal activity in *Si*, we compared the four GH18 chitinases encoded in the genome with respect to phylogenetic relationships, domain architecture, and transcriptional regulation. Of the four, only *Si*CHIT harbors a CBM5 domain, whereas the other three chitinases contain no additional domains besides the GH18 catalytic domain (Fig. 3A). *Si*CHIT and its closest paralog *Si*CHIT2 (PIIN_03542; Pirin1_74345) share the highest sequence identity (69.3%) and form a distinct phylogenetic pair among the *Si* GH18 chitinases (Fig. 3A; Supplementary Fig. 7). The catalytic DxxDxDxE motif located at the −1 subsite is conserved across all four chitinases, suggesting that each is catalytically active (Fig. 3B). Despite this shared motif, the chitinases exhibit distinct structural features. *Si*CHIT2 retains an 86-amino-acid C-terminal interdomain sequence (linker) but lacks a CBM5 domain. *Si*CHIT3 (PIIN_11727; Pirin1_71855)

lacks a predicted signal peptide, while *Si*CHIT4 (PIIN_07603; Pirin1_78411) features extended surface-exposed loops within the GH18 domain (Supplementary Table 1).

The evolutionary relationships among fungal GH18 chitinases, reconstructed using maximum-likelihood phylogenies of the catalytic domain (Supplementary Figs. 8 and 9), reveal three major fungal clades, including one enriched in CBM5 domain-containing (GH18–CBM5) chitinases. *Si*CHIT and *Si*CHIT2, which are tandem, adjacent genes in the *Si* genome, form a close sequence and structural pair distinct from other *Si* chitinases (Fig. 3C and Supplementary Figs. 8 and 9). Their high sequence identity and the absence of CBM5 in *Si*CHIT2 are consistent with a recent gene-duplication event followed by domain loss in one paralog. This supports a model in which the two genes may have undergone neo-functionalization or sub-functionalization within *Si*.

Across the broader fungal phylogeny, the GH18–CBM5-rich clade includes species with numerous paralogs, some of which also lack CBM5, consistent with multiple independent domain-loss events across Basidiomycota, or with the possibility that these patterns arose through tandem duplication followed by domain loss in an ancestral lineage and subsequent vertical inheritance. The observed lack of CBM5 is not uniquely associated with plant-associated lineages, indicating that domain loss is likely driven by diverse ecological pressures and functional requirements rather than lifestyle alone. The presence of the GH18–CBM5 domain combination in bacterial sequences that form an outgroup to this clade, together with the apparent absence of GH18–CBM5 chitinases in Ascomycota, raises the possibility that the CBM5-bearing GH18 lineage in Basidiomycota may have originated through horizontal gene transfer from bacteria, representing one plausible evolutionary scenario (Supplementary Figs. 8 and 9). Overall, this phylogenetic pattern supports a complex evolutionary history for GH18 chitinases involving gene duplication, domain gain and loss, and functional diversification.

To assess structural similarity between *Si*CHIT and *Si*CHIT2, we generated AlphaFold models[42,43] of both enzymes after removing the predicted signal peptides. The models confirmed the overall structural conservation of the GH18 catalytic domain and the presence of a C-terminal linker in *Si*CHIT2, but no CBM5 domain (Fig. 3D and E). The GH18 with the catalytic DxxDxDxE motif could also be confirmed in

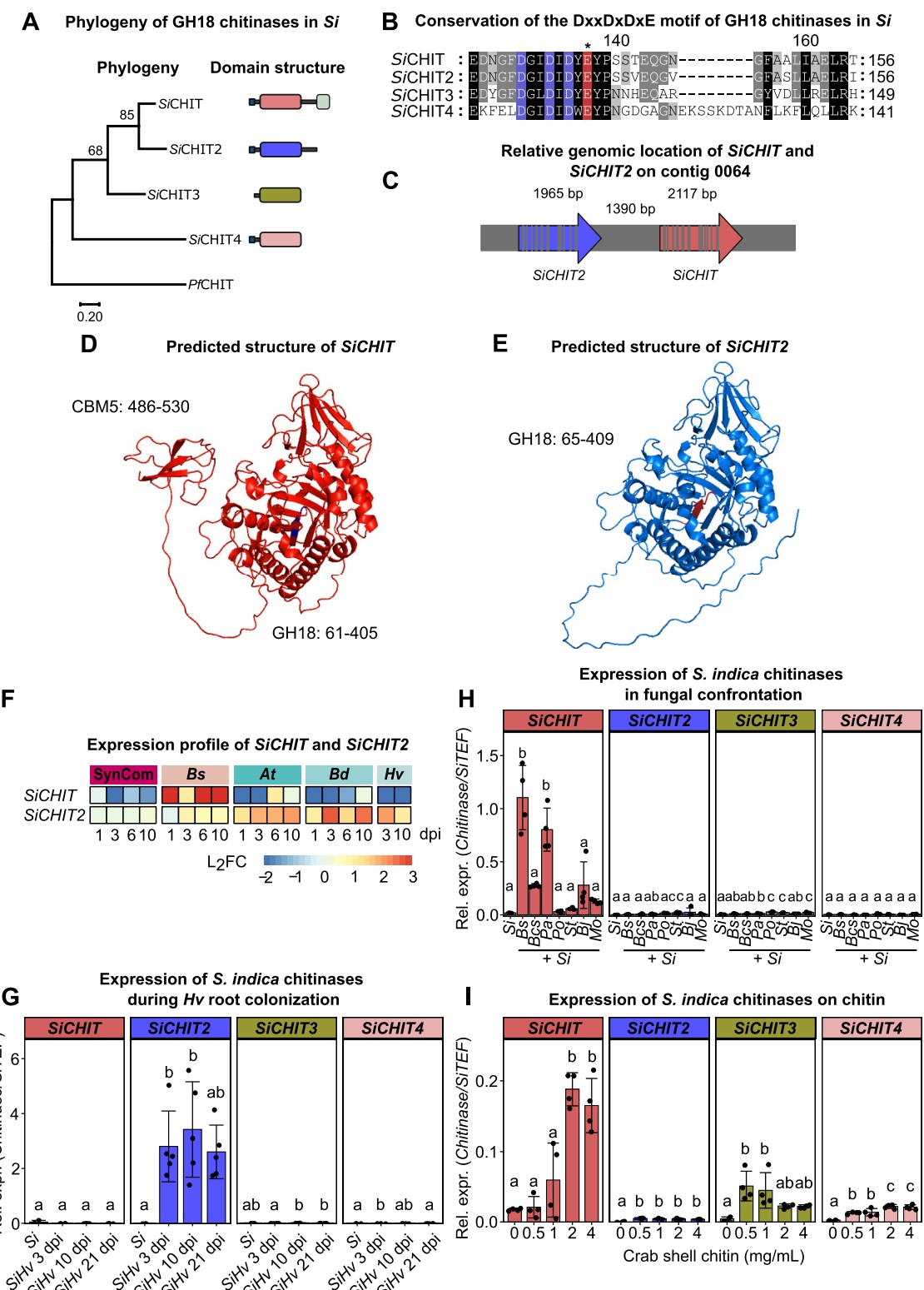

**A** Phylogeny of GH18 chitinases in *Si*

**B** Conservation of the DxxDxDxE motif of GH18 chitinases in *Si*

**C** Relative genomic location of *SiCHIT* and *SiCHIT2* on contig 0064

**D** Predicted structure of *SiCHIT*

**E** Predicted structure of *SiCHIT2*

**F** Expression profile of *SiCHIT* and *SiCHIT2*

**G** Expression of *S. indica* chitinases during *Hv* root colonization

**H** Expression of *S. indica* chitinases in fungal confrontation

**I** Expression of *S. indica* chitinases on chitin

*Si*CHIT3 and *Si*CHIT4, with the latter displaying three additional loop structures (Supplementary Fig. 10A, B).

To examine transcriptional regulation, we analyzed RNA-seq data from Eichfeld et al.[31] and found that *SiCHIT* and *SiCHIT2* were the only chitinases significantly upregulated in biotic contexts (Fig. 3F). *SiCHIT* was specifically induced during fungal confrontation, while *SiCHIT2* was induced during colonization of barley, *Brachypodium*, and

*Arabidopsis*. In contrast, *SiCHIT3* and *SiCHIT4* were not transcriptionally activated under any tested biotic conditions, including interactions with pathogenic fungi, beneficial bacteria, or different plant hosts (Fig. 3F, G). To extend the RNA-seq findings, we quantified the expression of all four GH18 chitinases in *Si* by qPCR following confrontation with *Bs* and six additional filamentous fungi representing the major fungal lineages Ascomycota, Basidiomycota, and

**Fig. 3 | *SiCHIT2* encodes a GH18 chitinase without CBM5 and is expressed during host plant colonization. A** Maximum-likelihood tree of all four *Si* GH18 chitinases. Signal peptides were predicted by SignalP 5.0 and removed prior to amino acid sequence alignment. Amino acid sequences were aligned using the MUSCLE algorithm. A maximum-likelihood tree was constructed with 500 bootstraps. A GH18 chitinase from *Plasmodium falciparum* was used as an outgroup. Symbols on the right indicate the domain architecture of the full-length proteins: small blue box: signal peptide; gray line: inter-domain regions; larger boxes (red, blue, green, beige): GH18 domain; small green box in *Si*CHIT: CBM5. The chitinase domain structures were created in BioRender (https://BioRender.com/l75bx0s). **B** Amino acid sequence alignment of the catalytic DxxDxDxE motif. Blue: conserved aspartate (D); red: conserved glutamate (E). The asterisk indicates the glutamate that was mutated to glutamin (Q) in the catalytically inactive *Si*CHIT$^{E196Q}$. **C** Relative location of *SiCHIT* and *SiCHIT2* in the genome of *Si*. Grey bars illustrate introns. *SiCHIT* has 11, and *SiCHIT2* has 10 introns. The scheme was created in BioRender (https://BioRender.com/l75bx0s). **D** and **E** Structure prediction of *Si*CHIT and *Si*CHIT2 by AlphaFold2 (predicted with ColabFold version 1.5.2). The sequences of predicted signal peptides were identified with SignalP 5.0 and removed prior to structure prediction. **F** Expression profile of *SiCHIT* and *SiCHIT2* during confrontation with a bacterial community (SynCom), the pathogen *B. sorokiniana* (*Bs*), or during colonization of the host plants *A. thaliana* (*At*), *B. distachyon* (*Bd*), or *H. vulgare* (*Hv*). Data was taken from Eichfeld et al. 2024[30]. Scale L$_2$FC refers to log$_2$-transformed normalized counts generated by DESeq2 from transcript counts. In *Hv*, transcript counts for *Hv* 1 and 6 dpi were removed from the analysis due to low sequence quality. **G** Expression of *Si* chitinases during *Hv* root colonization at 3, 10, and 21 dpi. Expression was inferred by relative expression of the fungal housekeeping gene *SiTEF* compared to *Si* chitinases by qPCR using the 2$^{-\Delta CT}$ method (mean ± SD, *n* = 3 for axenic *Si*; *n* = 5 for *Si* + plant samples). **H** Expression of *Si* chitinases during confrontation with different fungal species (*Bs* = *Bipolaris sorokiniana*; *Bcs* = *Brunneochlamydosporium* sp.; *Pa* = *Papulaspora equi*; *Po* = *Podospora* sp.; *St* = *Stachybotrys* sp.; *Bj* = *Bjerkandera* sp.; *Mo* = *Mortierella* sp.). Expression was inferred by relative expression of the fungal housekeeping gene *SiTEF* compared to *Si* chitinases by qPCR using the 2$^{-\Delta CT}$ method (mean ± SD, *n* = 4). **I** Expression of *Si* chitinases during growth on crab shell chitin-containing PNM medium at different concentrations. Expression was inferred by relative expression of the fungal housekeeping gene *SiTEF* compared to *Si* chitinases by qPCR using the 2$^{-\Delta CT}$ method (mean ± SD, *n* = 4). Statistical analysis: All displayed data (*n*) is derived from independent biological replicates. Different letters indicate significant differences according to one-way ANOVA followed by Tukey's HSD test (adjusted *p*-value < 0.05).

Mucoromycota. *SiCHIT* was induced in several of these interactions, whereas expression of *SiCHIT2*, *SiCHIT3*, and *SiCHIT4* remained low under all tested microbial conditions (Fig. 3H). When *Si* was grown on crab shell chitin as the sole carbon source, *SiCHIT*, *SiCHIT3*, and *SiCHIT4* were expressed at varying levels, whereas *SiCHIT2* expression remained comparatively low (Fig. 3I). Because these paralogs show distinct transcriptional profiles, we hypothesized that diversification in their upstream regulatory regions may underlie these differences. We therefore compared up to 500 bp upstream of each coding sequence using MEME Suite[44]. As a reference, we analyzed the promoter regions of six DELD effector genes, a known expanded gene family in *Si*[45,46]. Whereas DELD promoters displayed strong motif conservation, the putative regulatory regions of the four chitinases were highly divergent (Supplementary Fig. 11). Consistent with this divergence, weighted correlation network analysis (WCNA) shows that the four *Si* chitinases fall into distinct transcriptional modules: *SiCHIT* clusters in module 6, *SiCHIT2* in module 4, *SiCHIT3* in module 1, and *SiCHIT4* in the unassigned background module 0 (Supplementary Fig. 1A and Supplementary Table 1 in Brands et al.[47]). Notably, module 4 is plant-responsive and enriched for secreted proteins, including multiple predicted effectors such as DELDs, *Si*WSC3, *Si*FGB1, *Si*E5NT, and *Si*NucA, indicating that *SiCHIT2* is co-expressed with a host-induced secretome program.

Together, these findings suggest that *SiCHIT2* evolved a distinct function from *SiCHIT* following gene duplication. Their divergent domain architecture and transcriptional profiles are consistent with functional specialization, with *Si*CHIT adapted to fungal antagonism and *Si*CHIT2 to host-associated functions during colonization.

### Fusion of CBM5 reprograms *Si*CHIT2 for fungal antagonism

To test whether the CBM5 domain of *Si*CHIT is sufficient to confer antifungal activity, we constructed a chimeric protein consisting of the GH18 catalytic domain of *Si*CHIT2 fused to the proline-rich linker and CBM5 domain of *Si*CHIT (*Si*CHIT2$^{+CBM5}$). As a control, we generated a second construct in which only the linker was fused to the GH18 domain of *Si*CHIT2 (*Si*CHIT2$^{+link}$) (Fig. 4A). As observed for native *Si*CHIT and *Si*CHIT$^{-CBM5}$, the presence of CBM5 did not affect endpoint chitinase activity on crab shell chitin or chitin azure (Fig. 4B, C) but enhanced early hydrolysis (Supplementary Fig. 12). Notably, approximate Michaelis–Menten kinetics revealed that the presence of CBM5 reduced the maximal reaction velocity ($V_{max}$) while decreasing Km (Supplementary Fig. 13), indicating increased substrate affinity but a modest constraint on catalytic turnover under saturating substrate conditions. Together with the time-course assays on crab shell chitin and chitin azure (Fig. 1B, D; Supplementary Fig. 12), these findings indicate that CBM5 promotes substrate degradation during early reaction stages, while imposing steric limitations at substrate saturation. However, only *Si*CHIT2$^{+CBM5}$ negatively affected *Bs* spores, comparable to the effect of full-length *Si*CHIT, whereas *Si*CHIT2$^{+link}$ showed no inhibitory activity (Fig. 4D). This pattern was reflected in the hydrolytic activity on lyophilized *Bs* mycelium, where *Si*CHIT and *Si*CHIT2$^{+CBM5}$ showed strong activity (Supplementary Fig. 14A, B). Notably, *Si*CHIT$^{-CBM5}$ and *Si*CHIT2$^{+link}$ also displayed low activity on *Bs* mycelium. Chitinases lacking CBM5 may partially degrade the cell wall but appear unable to further hydrolyze cell wall fragments, as indicated by the non-migrating material observed in the TLC assay (Supplementary Fig. 14B). Consistent with these results, both *Si*CHIT and *Si*CHIT2$^{+CBM5}$ alleviated *Bs*-induced root growth inhibition in barley and reduced pathogen colonization, whereas *Si*CHIT$^{-CBM5}$ and *Si*CHIT2$^{+link}$ had no measurable protective effect (Fig. 4E, F).

These results demonstrate that fusing CBM5 to *Si*CHIT2 is sufficient to confer antifungal activity, indicating that CBM5 acts as a modular domain enabling functional adaptation toward microbial antagonism.

### *Si*CHIT2 suppresses chitin-triggered immunity and facilitates root colonization by *S. indica*

Given the distinct plant-induced expression profile of *SiCHIT2*, we hypothesized that this chitinase contributes to immune evasion during root colonization. Previous studies have shown that *Podosphaera xanthii* uses chitinase-like effectors to degrade immunogenic chitin oligomers and suppress host defense responses[48]. To test whether *Si* chitinases function similarly, we incubated immunogenic chitohexaose (A6) with recombinant *Si*CHIT, *Si*CHIT2, their respective variants, or the Ev control. The resulting hydrolysates were applied to barley roots to measure reactive oxygen species (ROS) production. ROS accumulation was completely abolished when chitohexaose was pretreated with either *Si*CHIT2$^{+link}$ or *Si*CHIT$^{-CBM5}$. In contrast, treatment with *Si*CHIT2$^{+CBM5}$ or full-length *Si*CHIT resulted in only partial or no suppression of ROS accumulation (Fig. 5A, B). An intermediate reduction in ROS accumulation was observed when chitohexaose was pre-treated with *Si*CHIT$^{E196Q}$, likely due to the inactive catalytic cleft sequestering a fraction of the chitohexaose. None of the recombinant chitinases induced ROS when applied in the absence of chitohexaose (Supplementary Fig. 15).

To assess whether immune evasion contributes to host colonization, we co-inoculated *Si* with 20 μM recombinant chitinase proteins and quantified fungal biomass in barley roots 3 days post-inoculation. *Si*CHIT$^{-CBM5}$ and *Si*CHIT2$^{+link}$ significantly enhanced root colonization, whereas the other chitinases did not (Fig. 5C). In bacterial GH19 and GH18 chitinases, removal of CBM5 has been shown to accelerate the

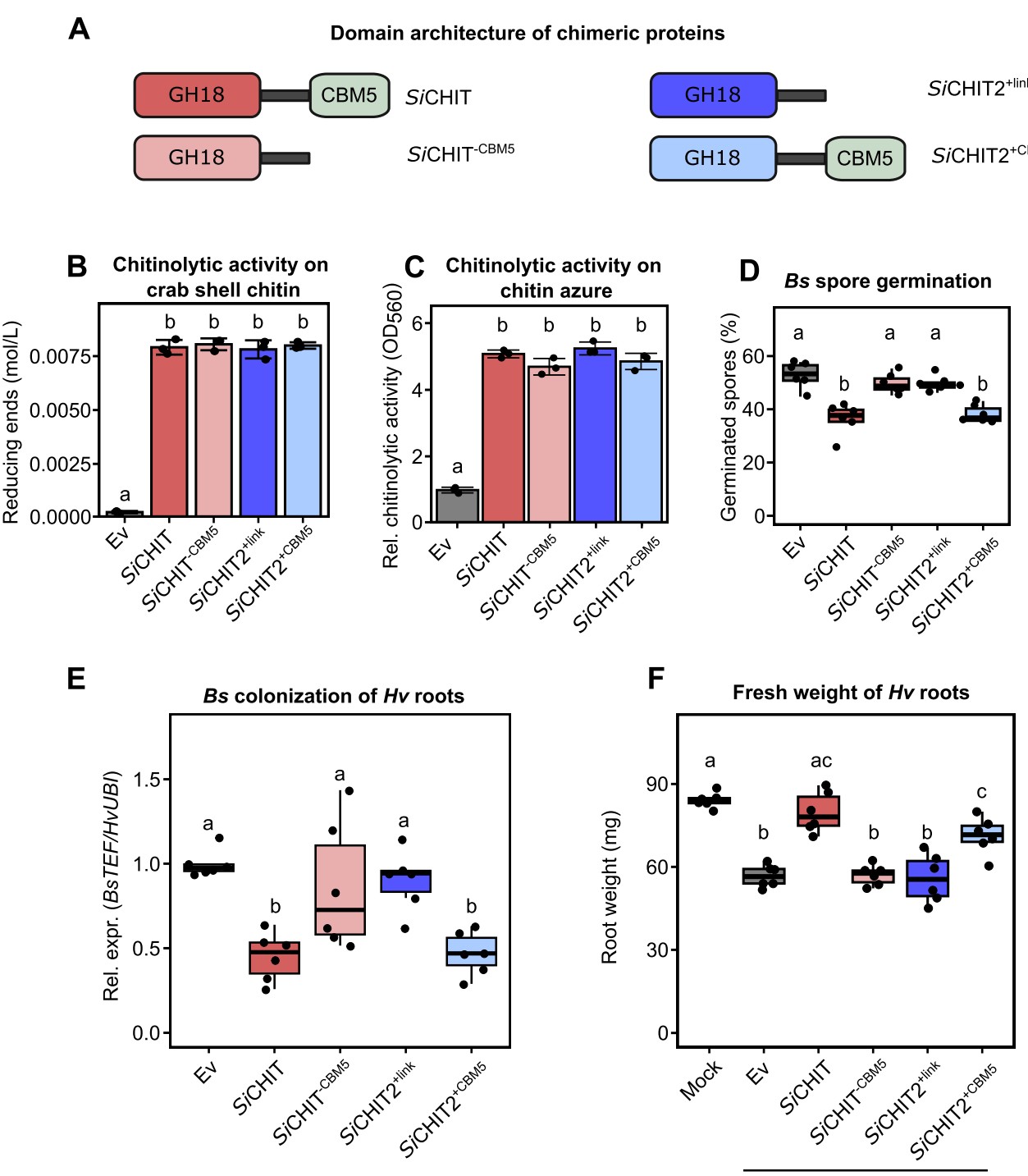

degradation of small, soluble chitooligomers[40,49], likely by reducing steric hindrance at the catalytic center. We therefore hypothesized that differences in the ability to evade chitin-triggered immunity might reflect differences in the degradation of soluble immunogenic chitooligosaccharides. To test this, we measured the chitinolytic activity of all recombinant variants on chitohexaose using a DNSA reducing-end assay. *Si*CHIT[−CBM5] and *Si*CHIT2[+link] showed higher release of reducing ends from chitohexaose compared to *Si*CHIT or *Si*CHIT2[+CBM5] (Supplementary Fig. 16A, B). In addition, time-resolved SEC-RI-MS analysis indicated that recombinant chitinase protein variants lacking the CBM5 domain depleted chitohexaose and the intermediate chitotetraose more rapidly, resulting in faster accumulation of shorter

oligomers (DP2-3) (Supplementary Fig. 16C, D). Because chitin-triggered ROS production is both threshold- and time-dependent, this kinetic bias provides a parsimonious explanation for the stronger suppression of immunity observed for these variants; however, additional *in planta* factors (e.g., protein stability/availability and solubility in the apoplast) may further modulate the magnitude of the response. Expression of the barley defense gene *PR10* was similar across treatments at 3 dpi despite increased colonization by *Si*CHIT[−CBM5] and *Si*CHIT2[+link] (Fig. 5D). This suggests that immune evasion mediated by recombinant *Si*CHIT[−CBM5] and *Si*CHIT2[+link] occurs primarily early during host contact, facilitating initial colonization. These results suggest that *Si*CHIT2's ability to evade ROS generation depends on the absence of

**Fig. 4 | Fusion of the CBM5 to *Si*CHIT2 activates its antifungal and plant-protective capacity. A** Domain architecture of recombinant proteins. *Si*CHIT2$^{+link}$ contains the GH18 domain of *Si*CHIT2 and the linker region of *Si*CHIT. *Si*CHIT2$^{+CBM5}$ contains the GH18 domain of *Si*CHIT2 and the linker region with the CBM5 of *Si*CHIT. The figure was created in BioRender (https://BioRender.com/l75bx0s). **B** Chitinase activity of recombinant chitinases on crab shell chitin at 10 μM and after 20 h at 28 °C. The relative amount of reducing ends was measured at 540 nm using a DNSA assay (mean ± SD, *n* = 3). **C** Chitinase activity of recombinant chitinases at 10 μM after 20 h at 28 °C on chitin azure (CA). Samples were spun down and supernatants were quantified at 560 nm and normalized to Ev (mean ± SD, *n* = 3). **D** Relative *Bs* spore germination 6 h post incubation in sterile chamber slides with Ev control or 10 μM of *Si*CHIT, *Si*CHIT$^{-CBM5}$, *Si*CHIT2$^{+link}$, or *Si*CHIT$^{+CBM5}$. Limits of the boxplots represent the 25th–75th percentile, the horizontal line represents the median, and the whiskers represent the minimum/maximum values without outliers (*n* = 6). **E** Colonization of *Hv* roots by *Bs* at three dpi inferred from relative expression of the fungal housekeeping gene *BsTEF* compared to the *Hv* housekeeping gene *HvUBI* by qPCR using the $2^{-\Delta CT}$ method. Colonization values were normalized to the Ev control. *Bs* spores were either pre-treated with the Ev control or 10 μM of recombinant chitinases for 20 h at 28 °C. Limits of the boxplots represent the 25th–75th percentile, the horizontal line represents the median, and the whiskers represent the minimum/maximum values without outliers (*n* = 6). **F** *Hv* root fresh weight at three dpi with *Bs* spores. *Bs* spores were either pre-treated with the Ev control or 10 μM of recombinant chitinases for 20 h at 28 °C. Limits of the boxplots represent the 25th–75th percentile, the horizontal line represents the median and the whiskers represent the minimum/maximum values without outliers (*n* = 6). Statistical analysis: All displayed data (*n*) is derived from independent biological replicates. Different letters indicate significant differences according to one-way ANOVA followed by Tukey's HSD test (adjusted *p*-value < 0.05).

CBM5, which may otherwise restrict access to soluble substrates and favor binding to insoluble chitin. This immune-evasive function, together with its host-specific transcriptional induction, underscores a distinct role for *Si*CHIT2 compared to its CBM5-containing paralog.

## Discussion

Effector proteins are central to the ecological success of root-colonizing fungi, mediating interactions not only with host plants but also with competing microbes. While traditionally associated with immune suppression[50], fungal effectors are now also recognized for their roles in inter-microbial competition. They facilitate the acquisition and maintenance of ecological niches in the rhizosphere and on root surfaces, thereby supporting fungal persistence and competitive advantage[11,16,31]. These dual roles are not restricted to pathogens; beneficial fungi such as *Si* employ similar strategies to persist in complex microbial communities, maintain host compatibility, and defend their niche. This functional versatility reflects broader evolutionary trends across the fungal kingdom, where transitions from saprotrophy to symbiosis are accompanied by the repurposing of ancestral traits to meet ecological demands.

Here, we show that *Si* encodes two closely related GH18 chitinases, *Si*CHIT and *Si*CHIT2, that have diverged in domain architecture and transcriptional regulation to perform distinct functions. Biochemically, our data indicate that the CBM5 primarily enhances substrate engagement and optimizes positioning of the GH18 catalytic domain on crystalline chitin, while preserving the enzyme's core chitinolytic activity. The presence or absence of CBM5 is sufficient to alter the efficiency and context of chitin hydrolysis in ways that yield distinct biological outcomes, ranging from microbial antagonism to enhanced suppression of chitin-triggered host immunity. Evolutionarily, this diversification could have arisen through gene duplication followed by domain gain or loss and transcriptional divergence, enabling modular specialization of a conserved chitinase scaffold. Consistent with this model, *Si*CHIT, which carries a C-terminal CBM5 domain, is induced during fungal confrontation, where it contributes to pathogen growth inhibition and plant protection. In contrast, *Si*CHIT2, which lacks CBM5, is induced during root colonization and more effectively suppresses chitin-triggered immune responses in the host (Fig. 6).

The importance of domain modularity in effector function is supported by studies in *Trichoderma* spp., where the addition of substrate-binding domains such as CBM1 or CBM18 increases their binding affinity to insoluble substrates and enhances antifungal activity by improving substrate targeting and proximity[37,38]. These findings suggest that the contributions of substrate-binding domains to antifungal capacity are not unique to CBM5 but may be a more general feature of different CBM families. However, a systematic analysis in which different CBMs are fused to the same GH18 catalytic domain to directly compare their contributions to antifungal activity is still lacking. In vivo, such effects can be difficult to disentangle in overexpression strains, because genome-wide redundancy among cell wall-degrading enzymes and antimicrobial effectors can mask the contribution of individual domains. Using purified proteins, we demonstrate that the naturally occurring CBM5 domain is critical for the antifungal activity observed for *Si*CHIT. Its deletion affects fungal antagonism, and its introduction into *Si*CHIT2 is sufficient to restore this function. This is consistent with studies in bacterial chitinases, where CBM5 loss reduces antifungal activity, likely due to impaired substrate binding[40,51]. In our assays, CBM5 enhanced early hydrolysis rates and binding to crystalline chitin as well as fungal cell walls and lowered $K_m$, but did not affect the product profile. This suggests a proximity-based enhancement mechanism, in which CBM5 anchors the enzyme at the fungal cell wall.

Unlike antimicrobial or agglutinating lectins such as WSC3, a β-1,3-glucan-binding lectin from *S. indica* that agglutinates fungal cells independently of enzymatic activity[52], CBM5 alone does not provide antifungal activity; the GH18 catalytic core is required. Some fungal pathogens employ catalytically inactive chitinase effectors or active chitinases to suppress host immunity by sequestering or degrading free chitin oligomers[36,48,53,54]. Similarly, we show that both *Si*CHIT2 and *Si*CHIT$^{-CBM5}$ can suppress chitohexaose-induced ROS production. These findings suggest that CBM5 may hinder activity on soluble substrates not by altering catalytic specificity, but by limiting access to short oligomers through spatial constraints. CBM5 domains are structurally optimized for engaging extended, crystalline chitin surfaces via aromatic stacking and hydrogen bonding[55,56]. Accordingly, CBM5-containing chitinases like *Si*CHIT likely localize to insoluble polysaccharides in fungal cell walls, anchoring the enzyme to non-immunogenic material. Although such substrate targeting could in principle restrict access to soluble MAMPs, this is unlikely in the ROS assay, where only chitohexaose is present. A more plausible explanation is that CBM5 increases the rigidity or steric bulk of the protein, thereby reducing flexibility and limiting productive interaction with short ligands. In contrast, CBM5-lacking variants such as *Si*CHIT2 remain free to access and degrade free immunogenic chitin oligomers. This aligns with the fact that truncation of the CBM5 of bacterial GH18 and GH19 chitinases increases their ability to degrade short and soluble chitooligosaccharides at the cost of degradation of crystalline chitin[40,49]. This suggests that suppression of host immunity may be favored by a CBM5-free configuration optimized for apoplastic MAMP degradation during root colonization. Our analysis of the chitinolytic activity of the different recombinant proteins on immunogenic chitohexaose underscores this hypothesis. Such structural neofunctionalization provides a plausible trajectory for the evolution of fungal effector functions. Given the saprotrophic ancestry and facultative symbiotic lifestyle of *Si*, the antimicrobial function of *Si*CHIT likely evolved under selective pressure from microbial competitors, whereas *Si*CHIT2 may have emerged as an immune-suppressive effector during the transition to a root-associated niche.

The functions of the other two predicted GH18 chitinases in *Si* remain unknown. Because *Si*CHIT3 lacks a predicted signal peptide for

secretion, it is likely to function intracellularly. In *Histoplasma capsulatum*, the chitinase Cts3, also lacking a signal peptide, has been implicated in cell division, septum formation, and cell wall remodeling[57], suggesting that *Si*CHIT3 may fulfill comparable intracellular roles. However, the induction of *SiCHIT3* during growth on chitin indicates that its activity could also be modulated by

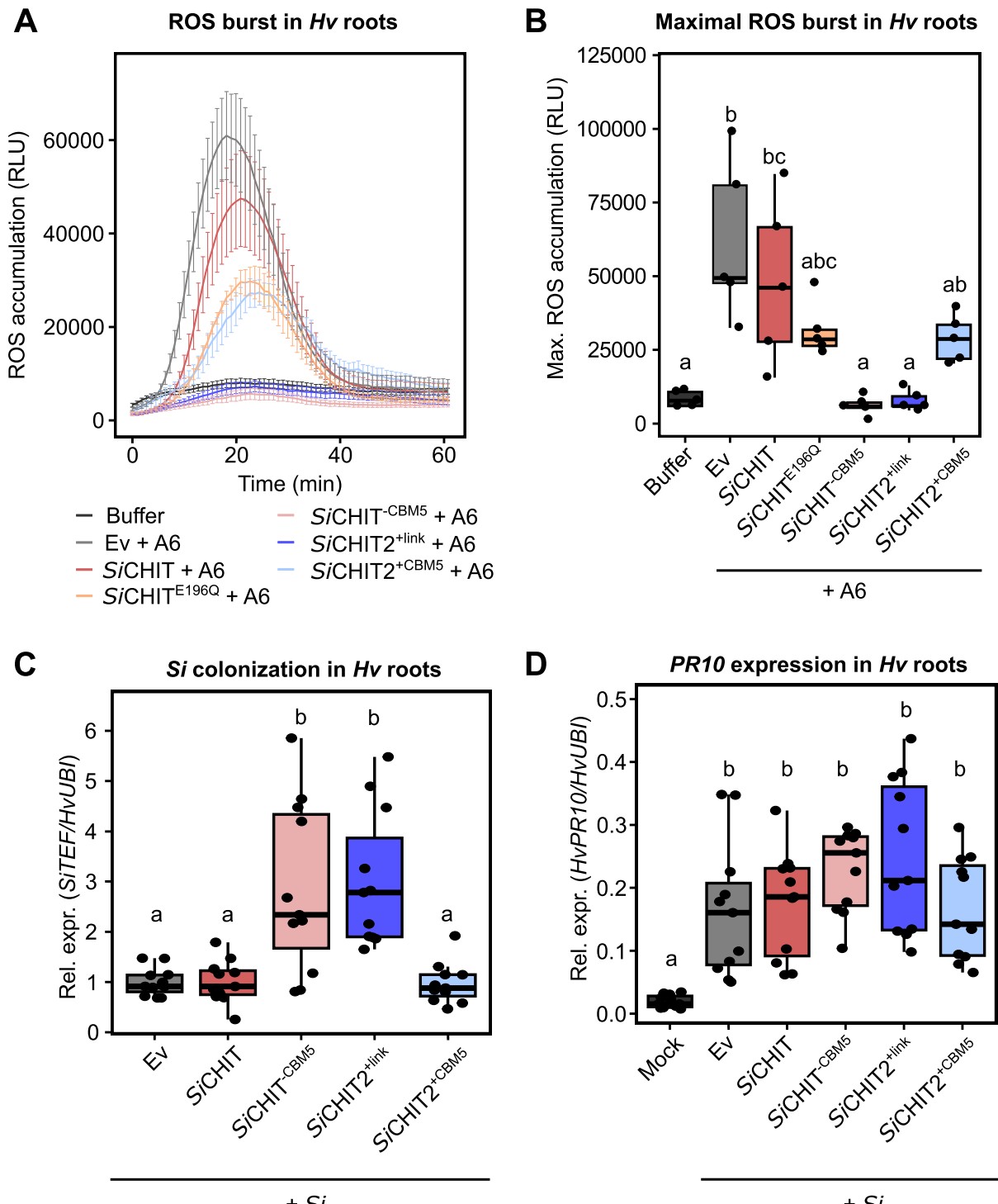

**Fig. 5 | *Si*CHIT2 suppresses chitin-triggered immunity and facilitates root colonization by *Si* in *Hv*. A** ROS burst of 4 d *Hv* roots after treatment with 125 nM chitohexaose (A6). Chitohexaose was incubated with the indicated recombinant chitinases in an equimolar concentration (10 μM) 20 h prior to treatment of *Hv* roots (mean ± SEM, *n* = 5). **B** Maximum values of ROS accumulation of *Hv* roots (data from **A**) (*n* = 5). **C** Colonization of *Hv* roots by *Si* at three dpi inferred from relative expression of the fungal housekeeping gene *SiTEF* compared to the *Hv* housekeeping gene *HvUBI* by qPCR using the $2^{-\Delta CT}$ method. Colonization values were normalized to the Ev control. *Si* spores were germinated for 16 h at 28 °C and either incubated with the Ev control or 20 μM of recombinant chitinases immediately before *Hv* root inoculation (*n* = 11). **D** Transcript accumulation of the *Hv* defense marker gene *HvPR10* at three dpi inferred by relative expression of *HvPR10* relative to the *Hv* housekeeping gene *HvUBI* by qPCR using the $2^{-\Delta CT}$ method (*n* = 11). Limits of the boxplots represent the 25th–75th percentile, the horizontal line represents the median, and the whiskers represent the minimum/maximum values without outliers. Statistical analysis: All displayed data (*n*) is derived from independent biological replicates. Different letters indicate significant differences according to one-way ANOVA followed by Tukey's HSD (adjusted *p*-value < 0.05).

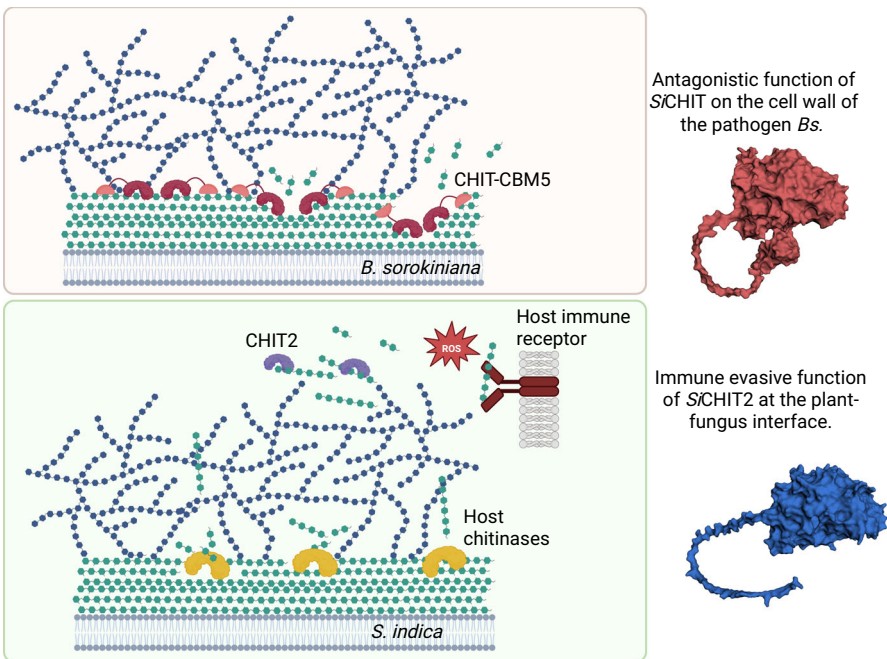

**Fig. 6 | Model displaying the divergent functions of *Si*CHIT and *Si*CHIT2.** During confrontation with the pathogen *Bs*, *Si* secretes a GH18-CBM5 chitinase that efficiently inhibits the growth of *Bs*, enabling *Si* to secure its ecological niche. To effectively function in this context, *Si*CHIT relies on the presence of the CBM5 domain that contributes to targeted fungal cell wall degradation. The paralogous

*Si*CHIT2 has a catalytically active GH18 domain but no CBM5 domain. It serves *Si* during host colonization by preventing the elicitation of chitin-triggered immunity at the fungus–plant interface, putatively by degrading immunogenic chitin oligomers that are released from the cell wall of *Si* by host chitinases. Figure was created in BioRender (https://BioRender.com/w0gv9yi).

environmental or external cues. By contrast, *Si*CHIT4 contains a predicted signal peptide and is expressed during growth on chitin, consistent with a function in extracellular chitin hydrolysis. Notably, *Si*CHIT4 possesses three loop regions within its GH18 catalytic domain. Recent studies of plant GH19 chitinases showed that combinations of chitinolytic activity and extended loop regions can enhance antifungal activity by increasing binding to fungal cell walls[58,59]. Whether such loop structures serve as an alternative to CBMs for mediating antifungal activity in microbial chitinases remains an open question. In our study, *Si*CHIT4 was not induced during confrontation with other fungi, arguing against a major role in fungal competition under the conditions tested. A broader transcriptional survey using an expanded panel of fungal competitors may help clarify whether *Si*CHIT4 participates in inter-fungal interactions under specific ecological contexts.

Another important question is how *Si* protects itself against *Si*CHIT activity during inter-fungal encounters. Similar concerns have been raised for the mycoparasitic *Trichoderma* species[60]. It has been proposed that co-expressed lectins with cell-wall binding capabilities may shield the producer's own cell wall from degradation. Such lectins could confer conditional resistance to autolysis, thereby allowing chitinases to perform alternative roles during fungal growth and development[60]. Notably, *Si* expresses several lectins with predicted cell wall-binding functions during confrontation with *Bs*[31]. However, experimental evidence supporting this protective mechanism remains lacking.

The functional divergence between *Si*CHIT and *Si*CHIT2 exemplifies how domain loss and regulatory rewiring can drive distinct biological outcomes without altering the core chitinase activity. By comparison, the *Verticillium dahliae* effector *Vd*AMP3 illustrates a different evolutionary strategy: co-opted from an ancient antimicrobial peptide, it functions in plant microbiota manipulation[14]. Rather than requiring extensive structural innovation and transcriptional rewiring, *Vd*AMP3 shows how effectors can be repurposed for new ecological contexts, highlighting the diversity of evolutionary routes by which fungal effectors adapt to host environments.

Together, our findings show that domain gain or loss can reprogram effector proteins for distinct ecological functions, enabling mutualistic fungi to balance rhizosphere competition with host interactions.

## Methods

### Plant and fungal materials

*Hordeum vulgare* (*Hv*, L. cv Golden Promise) and *Arabidopsis thaliana* (*At*, Col-0) were used as plant hosts. *Hv* seeds were surface-sterilized in 6% sodium hypochlorite for 1 h under constant shaking and subsequently washed five times for 30 min in sterile water. Seeds were germinated for 4 days on wet filter paper at 21 °C in the dark. Seeds of *At* were surface-sterilized in 70% ethanol for 10 min and 100% ethanol for 7 min. *At* seeds were placed on ½ MS medium (Murashige-Skoog medium including vitamins, pH 5.6) containing 1% sucrose and stratified for 3 days in the dark and 4 °C prior to germination on a day–night cycle of 8/16 h at 22/18 °C, 60% humidity, and 125 µmol/m² light intensity for 7 days. *Serendipita indica* (*Si*, DSM11827) and *Bipolaris sorokiniana* (*Bs*, ND90Pr) were used as fungal species. *Si* was grown on CM medium (Complete Medium, 1.5% agar[61]) at 28 °C for 28 days in the dark. *Si* spores were harvested by adding sterile water and scraping the spores with a scalpel. The suspension was filtered through a sterile Miracloth filter and spores were centrifuged at 2000×*g*. The supernatant was discarded and spores were washed three times in water. *Bs* was grown on modified CM medium[25] at 28 °C for 14 days in the dark. *Bs* spores were harvested like *Si* spores, but a Drigalski spatula was used to scrape the spores instead of a scalpel.

### Preparation of fungal inoculants and plant colonization

For the preparation of *Bs* inoculants, spores were harvested from 21-day-old plates and incubated with the respective recombinant proteins (10 µM in 50 mM phosphate buffer) for 16 h at 28 °C in the dark. For the preparation of *Si* inoculants, spores were harvested from 28-day-old plates and allowed to germinate overnight in sterile water at 28 °C. Prior to plant root inoculation, the respective recombinant proteins

were added to 20 µM in 50 mM phosphate buffer. Spore mixtures were diluted to 5000 spores/mL for *Bs* and 500,000 spores/mL for *Si* to yield the final working inoculants. For inoculation of *At*, ten sterile 7-day-old seedlings were placed on solid 1/10 PNM (pH 5.6) plates (0.005% $KNO_3$, 0.005% $KH_2PO_4$, 0.0025% $K_2HPO_4$, 0.049% $MgSO_4$, 0.00472% $Ca(NO_3)_2$, 0.0025% NaCl, 0.5% (v/v) Fe-EDTA stock solution, 1.2% Gelrite, pH 5.6. After autoclaving, 10 mL 1 M MES pH 5.6 was added. The EDTA stock solution contains 2.78% $FeSO_4 \times 7\ H_2O$, 4.13% $Na_2EDTA \times 2H_2O$). Concentrations are indicated in w/v if not stated otherwise. The roots were treated with 1 mL of the respective inoculant and grown on a day–night cycle of 8/16 h at 22/18 °C, 60% humidity and 125 µmol/m² light intensity for 3 days. The root elongation was quantified by measuring the difference in root length at the start of the inoculation and the end of the experiment at 3 days post-inoculation (dpi). Roots were washed with water and snap frozen in liquid nitrogen for further RNA extraction. For inoculation of *Hv*, four sterile seedlings were placed in glass jars containing solid 1/10 PNM medium (pH 5.6) and inoculated with 3 mL of the respective inoculant. The inoculated plants were grown on a day–night cycle of 16/8 h at 22/18 °C and 60% humidity and a light intensity of 108 µmol/m² for 3 days. Root weight was quantified at 3 dpi. Roots were washed in water and snap frozen for further RNA extraction.

### Preparation of fungal confrontations
For the preparation of fungal inoculants, liquid CM medium was inoculated with 1 mL of *S. indica* spores at a concentration of 500,000 spores/mL or 1 mL of *B. sorokiniana* spores at a concentration of 5000 spores/mL. For all other fungi, mycelium was scraped from plates and transferred to liquid CM. All fungi were grown for 6 days at 28 °C and 120 rpm. The mycelium was blended and re-suspended in fresh CM[61] and regenerated for another 24 h. Regenerated mycelium was filtered through Miracloth filters and washed thoroughly with sterile MilliQ water. Fungal mycelia were mixed in a 1:1 ratio, and 1 g of mycelium was streaked on PNM plates containing 15% agar. After 6 days, the mycelia were harvested, dried with tissue papers, and snap frozen in liquid nitrogen.

### Antifungal activity assay
To quantify the *Bs* spore germination, isolated spores were resolved in liquid CM to a concentration of 100,000 spores/mL and 100 µL were added to the wells of sterile Lab-Tek 2 chamber slides with cover (VWR). 100 µL of the respective treatment (Ev control or purified protein) was added to a final concentration of 10 µM. The germination rate of spores was quantified 6 h after treatment.

### RNA extraction from plant roots and qPCR
Plant materials were ground in liquid nitrogen, and 1 mL of TRIzol reagent (Thermo Fischer Scientific) was added. Samples were vortexed for 20 s until all material was resolved. 200 µL chloroform was added, and samples were vortexed again. Samples were centrifuged at 17,000×*g*, and 4 °C for 30 min and 500 µL of the upper aqueous phase was transferred to 500 µL isopropanol. The nucleic acids were precipitated at −20 °C for 16 h and pelleted by centrifugation at 17,000×*g* and 4 °C for 30 min. Pellets were washed three times with 70% ethanol diluted in DEPC water. Remaining DNA was removed by adding 1 µL DNase1 (1 U/µL) in 3 µL DNase1 buffer (10x) (Thermo Fischer Scientific) and 26 µL RNase-free water. Samples were incubated at 37 °C for 30 min. DNase digest was stopped by heating to 70 °C for 5 min and subsequent precipitation with 2 volumes of isopropanol and ½ volume of 7.5 M ammonium acetate for 16 h at −20 °C. Samples were washed twice with 70% ethanol in DEPC water and resolved in 30 µL RNase-free water. Concentrations were determined using a NanoDrop 2000c spectrophotometer (Thermo Fischer Scientific).

1 µg of RNA was used for reverse transcription into cDNA (Thermo Fischer Scientific). First, 1 µL of random hexamer and of oligo-dt oligomers were added and incubated at 65 °C for 5 min and subsequently placed on ice. 4 µL 5× reaction buffer, 1 µL Riboblock RNase inhibitor (20 U/µL), 2 µL 10 mM dNTPs, and 2 µL MMLV reverse transcriptase were added, and samples were incubated at 42 °C for 1 h. cDNA synthesis was stopped by heating to 70 °C for 5 min. qPCR was conducted using the primers listed in Supplementary Table S2.

### Cloning of chitinase expression vectors
The coding sequence (CDS) of the chitinase genes was amplified by PCR with Q5 High-Fidelity Polymerase (New England Biolabs) using the primers listed in Supplementary Table 2. PCR products were cloned using the Gibson assembly cloning technique into the PQE80L expression vector (Qiagen, Hilden, Germany) after vector linearization with Kpn1 (New England Biolabs). For the production of *Si*CHIT2⁺link and *Si*CHIT2⁺CBM5, the inter-domain region (linker) or the inter-domain region with the CBM5 domain of *Si*CHIT2 was fused to the C-terminus of the GH18 domain of *Si*CHIT2.

### Production and purification of recombinant protein in *E. coli*
Chemically competent *E. coli* BL21 cells were transformed with the vectors for protein expression under control of a lac operon. 5 mL starter cultures were prepared in liquid LB medium containing 100 µg/mL carbenicillin and grown for 16 h at 37 °C and 180 rpm. Starter cultures were transferred to 300 mL of LB medium with carbenicillin and grown to an $OD_{600}$ of 0.5 at 28 °C. Gene expression was induced by adding IPTG to a final concentration of 1 mM and cells were grown at 16 °C for 20 h. Cells were pelleted by centrifugation at 5000×*g* for 15 min at 4 °C. Bacterial pellets were resolved in 5 mL lysis buffer (50 mM $NaH_2PO_4$, 300 mM NaCl, and 10 mM imidazole, pH 8) and sonicated three times for 1 min with pulses on ice and centrifuged at 16,200×*g* for 30 min. 5 mL of the supernatant was added to 1 mL of Nickel-NTA Agarose (Qiagen, Hilden, Germany) and incubated under continuous shaking at 4 °C for 1 h. Samples were spun down at 500×*g* for 10 s, the flow through was removed, 4 mL wash buffer (50 mM $NaH_2PO_4$, 300 mM NaCl, and 20 mM imidazole, pH 8) was added, and samples were incubated under continuous shaking at 4 °C for 10 min. Washing steps were repeated five times. For elution, 1 mL of elution buffer (50 mM $NaH_2PO_4$, 300 mM NaCl, and 250 mM imidazole, pH 8) was added, and samples were incubated under continuous shaking at 4 °C for 10 min. Eluted fractions were dialyzed against 50 mM phosphate buffer (pH 6) overnight, and protein concentrations were measured using Bradford reagent. To test the purity of the protein samples, aliquots were boiled for 5 min in SDS-sample buffer, loaded on a 10% SDS gel, and visualized by Coomassie staining.

### Chitinase activity assays
To quantify the enzymatic activity of the purified chitinases, 2 mg/mL chitin azure (Sigma Aldrich; C3020) was incubated with 5 µM recombinant chitinase in a total volume of 200 µL. Reactions were incubated at 28 °C for 20 h. Reactions were stopped by heating to 95 °C for 5 min, and samples were centrifuged 5 min at 17,000×*g* and the absorbance of supernatants was quantified at 560 nm. For quantifying reducing ends, 20 mg crab shell chitin (Sigma Aldrich; C7170) or >75% de-acetylated chitosan from shrimp cells (Sigma Aldrich; C3646, average molecular weight -360 kDa) were incubated with 10 µM enzyme in 200 µL 50 mM phosphate buffer for 20 h at 28 °C. Reactions were stopped by heating to 95 °C for 5 min. Samples were centrifuged for 5 min at 17,000×*g*, and the supernatant was mixed in a 1:1 ratio with DNSA (3,5-dinitrosalicylic acid) reagent. DNSA reagent is prepared by mixing 1% (w/v) 3,5-dinitroslicylic acid, 1.6% (w/v) sodium hydroxide, 30% (w/v) potassium sodium tartrate tetrahydrate. The samples were heated for 10 min at 95 °C and subsequently transferred to ice. The absorbance of the samples was measured at 540 nm. $OD_{540}$ values

were converted to the concentration of reducing ends (mol/L) using a N-acetylglucosamin standard series.

For time course experiments with CA, crab shell chitin and chitosan, samples were prepared as described above in a final volume of 1 mL and placed in a 28 °C incubator, allowing reaction tubes to shake horizontally. 150 μL of the samples was taken every hour.

For the kinetic analysis, crab shell chitin at concentrations from 10 to 120 mg/mL was incubated with 5 μM recombinant enzyme for 60 min. Curve fitting and Michaelis−Menten parameters were calculated using the package "renz" and "drc" in R.

For TLC on *Bs* mycelium, 15 mg lyophilized and washed mycelium were weighed in 2 mL reaction tubes. For washing, lyophilized mycelium was heated at 80 °C for 5 min in 1 mL 50 mM phosphate buffer and then washed in 50 mM phosphate buffer until supernatants did not color Bradford dye (5 times). The recombinant chitinases were added to a final concentration of 5 μM in 200 μL. Reactions were incubated at 28 °C for 20 h and stopped by heating to 95 °C for 5 min. Samples were centrifuged at 17,000×*g* for 5 min. 10 μL of the supernatant was separated on 60 $F_{254}$ TLC (Sigma Aldrich) silica gel plates with running buffer (n-butanol/methanol/28% ammonium hydroxide/water, 5/4/2/1). Plates were dried and then bathed in developer solution (1.6 g diphenylamine/1.6 mL aniline/12 mL 85% phosphoric acid/80 mL acetone). TLC plates were developed in an incubation oven at 100 °C for 20−30 min.

### Chitinase subsite specificity and product profiles

The subsite specificity of *Si*CHIT[-CBM5] was determined as previously reported[31,62] in biological triplicates. *Si*CHIT[-CBM5] was used at a final concentration of 0.2 μM on 1 mg/mL of chitosans with an average degree of acetylation (DA) of 10%, 35%, or 50% and an average degree of polymerization (DP) around 1000 in 50 mM of sodium acetate buffer. Incubation was performed for 1 h or 24 h. Oligomers produced by *Si*CHIT[-CBM5] were N-acetylated using [$^2H_6$]acetic anhydride, and MS[1] samples were subsequently mixed with internal chitin standards with DPs from 1-6, which are double-isotopically labeled via N-acetylation with [$^2H_6$; $^{13}C_4$]acetic anhydride. These were analyzed by HILIC-ESI-MS[1] using a Waters ACQUITY Premier UPLC System coupled to a Waters Synapt XS HDMS 4k mass spectrometer. Oligomer mixtures were separated using an Acquity UPLC BEH Amide column (1.7 μm, 2.1 mm × 50 mm; Waters Corporation, Milford, MA, USA) and a Van-Guard precolumn (1.7 μm, 2.1 mm × 5 mm; Waters Corporation, Milford, MA, USA). The flow rate was set to 0.4 mL/min, the column oven temperature was set to 40 °C, and 2 μL of the sample, including 50 ng of each of the internal standards (double isotopic labeled chitin oligomers of DP 1−6), was injected into the system. The sample was separated using an LC run lasting 10.5 min, with the following gradient elution profile: 100% A (80:20 acetonitrile:$H_2O$ with 10 mM $NH_4HCO_2$ and 0.1% (v/v) HCOOH) for 0.0−1.0 min, followed by a linear gradient to 20% (v/v) B (20:80 acetonitrile:$H_2O$ with 10 mM $NH_4HCO_2$ and 0.1% (v/v) HCOOH) from 1.0-7.5 min, and then to 75% (v/v) B from 7.5 to 8.5 min. From 8.5 to 9.0 min, the flow was isocratic 75% (v/v) B, followed by a linear gradient from 25% (v/v) A to 100% A from 9.0 to 9.2 min, and an isocratic elution of 100% A from 9.2 to 10.5 min. The MS[1] measurements were conducted in positive resolution mode with normal dynamic range sensitivity, with a mass range of *m/z* 50−2000 and a scan time of 0.25 s. To improve mass accuracy, lock spray calibration was performed using leucine encephalin, injected in 10-s intervals for one second as a lock mass. The capillary voltage of the source was set to 3 kV, the sampling cone to 35, the source offset to 20, while the source and desolvation temperatures were set to 80 and 250 °C, respectively. Cone gas flow was set to 0 l/h, and the desolvation gas flow was set to 500 l/h. The nebulizer pressure was set to 5.8 bar. Quantitative pattern determination by MS[2] was performed following the method previously described[62]. Briefly, the oligomers produced by *Si*CHIT[-CBM5] after N-acetylation using [$^2H_6$]acetic anhydride as

described above were incubated in $H_2^{18}O$ overnight to label the reducing end of the products with $^{18}O$ before MS[2] analysis. The Waters LC−MS system used for the MS[1] analysis was also used for this analysis. The TOF MS/MS mode was employed to determine the pattern of the N-acetylated and $^{18}O$-labeled oligomers. Each oligomer containing at least one natural and one isotopically labeled acetyl group was individually isolated using the quadrupole of the MS system with an LM resolution of 15 to enable the isolation and fragmentation of monoisotopic peaks. The collision energy in the trap was set to 18 V for dimers, 20 V for trimers, 22 V for tetramers, 24 V for pentamers, and 26 V for hexameric products. MS data was analyzed using MassLynx V4.2 (Waters, USA) and the intensities of the MS signals for each oligomer or fragment were quantified using Skyline 23.1 (MacCoss Lab Software, University of Washington, USA[63]). Exemplary mass spectra are shown in Supplementary Fig. 4. The MS[1] signal intensities of each product were compared to the intensity of the corresponding double isotopic-labeled standard to calculate the amount of each product based on the 50 ng of injected standard. The MS[2] signal intensities of all y-fragments and the largest b-fragment were used to calculate the proportions of the different patterns of the oligomers[64]. The combination of the resulting datasets allows for calculating the amount of each product of defined DP, DA, and pattern, which can in turn be used to quantify the frequency of acetylated units at the −2 to +2 subsites of *Si*CHIT[-CBM5].

To determine product profiles, 1 mg of each insoluble substrate was incubated with the corresponding enzyme (5 μM) in 40 μL of ammonium acetate buffer (50 mM, pH 5) for four days at 37 °C and 180 rpm. Subsequently, the samples were filtered through centrifugal filters (modified PES, 3 kDa cutoff; VWR, Germany), dried in a vacuum concentrator at 30 °C, and resuspended in 10 μL MilliQ water. A volume of 2 μL was analyzed using the HILIC-ESI-MS[1] system described above in sensitivity mode. Different from subsite specificity analyses, the column temperature was set to 35 °C, and the following gradient elution profile was used: 100% A for 0.0−2.0 min, followed by a linear gradient to 75% (v/v) B from 2.0 to 5.0 min. From 5.0 to 8.0 min, the flow was isocratic 75 % (v/v) B, followed by a linear gradient from 25% (v/v) A to 100% A from 8.0 to 8.2 min, and an isocratic elution of 100% A from 8.2 to 10.0 min. Resulting data was analyzed using MassLynx V4.2 (Waters, USA) and the intensities of the MS signals for each oligomer were quantified using Skyline 23.1 (MacCoss Lab Software, University of Washington, USA[63]) as described above, exemplary mass spectra are shown in Supplementary Fig. 2.

### SEC−RI−MS time-resolved product profiling on chitohexaose

Product profiles generated by the five enzymes during hydrolysis of the chitin hexamer (A6) were monitored by live sampling of ongoing reactions coupled to size exclusion chromatography with refractive index and mass spectrometric detection (SEC−RI−MS), as described previously[65]. Briefly, A6 (0.5 mg/mL) was prepared in ammonium acetate buffer (150 mM, pH 4.5 adjusted with acetic acid) and preheated to 37 °C. Reactions were assembled directly in the autosampler by mixing 20 μL substrate solution with 10 μL preheated enzyme stock (0.3 μM in the same buffer), yielding final concentrations of 0.33 mg/mL A6 and 0.10 μM enzyme in a total volume of 30 μL. Immediately after mixing, 3 μL were withdrawn and injected as the first timepoint (1.5 min). Subsequent 3 μL injections were taken at 17.0, 32.5, and 48.0 min (i.e., every 15.5 min after the initial 1.5 min timepoint) while incubating at 37 °C in the autosampler. MS and RI data were exported in DataAnalysis 4.1 (Bruker) and processed in Skyline 23.1 (MacCoss Lab Software)[63] and OriginPro 2025 (OriginLab), respectively.

### Substrate binding assay

15 mg of crab shell chitin (Sigma Aldrich; C7170), Chitosan (Sigma Aldrich; C3646) or lyophilized mycelium of *Bs* were weighed in a 2 mL

reaction tube. For washing, lyophilized mycelium was heated at 80 °C for 5 min in 1 mL 50 mM phosphate buffer and then washed in 50 mM phosphate buffer until supernatants did not color Bradford dye (5 times). Samples were incubated with 400 μL chitin-binding buffer (2 mM $KH_2PO_4$, 8 mM $Na_2HPO_4$, 2 mM KCL) and 100 μL of 20 μM recombinant protein, shaking at 4 °C for 1 h. Samples were centrifuged at 17,000×$g$ for 5 min and 60 μL of the supernatant was mixed with 60 μL Bradford reagent. Protein amounts were quantified at 595 nm and compared to the input amount. In addition, supernatants or pellets were boiled in SDS-sample buffer at 95 °C and separated via SDS–PAGE on a 10% SDS gel and stained with Coomassie brilliant blue.

### Quantification of reactive oxygen species (ROS)

For elicitor preparation, 10 μM of chitohexaose was incubated with 10 μM of recombinant chitinase or the buffer control for 20 h at 28 °C. Reactions were stopped by heating to 95 °C for 5 min and chitohexaose was diluted to 250 nM.

Four root pieces (0.5 cm) of 4-day-old barley seedlings were transferred to each well of a 96-well microtiter plate (white, flat bottom) containing 200 μL of 2.5 mM 2-(N-morpholino)ethanesulfonic acid (MES) buffer, pH 5.6. The plate was incubated for 16 h at 21 °C for recovery. The buffer was replaced with 100 μL 2.5 mM MES buffer containing 20 μM LO-12 (Wako Chemicals) and 20 μg/mL Horseradish peroxidase (HRP) (Sigma-Aldrich). After 25 min incubation in the dark, 100 μL of 250 nM chitohexaose elicitor solution was added to each well. Chemiluminescence was measured using a TECAN SPARK 10 M microplate reader over all wells for 2 h with an integration time of 450 ms.

### Fungal sequence extraction and GH18 domain extraction

Fungal sequences were manually extracted from JGI Mycocosm portal[66] (Supplementary File 1) and only species encoding GH18-CBM5 were included. Basidiomycetes and Ascomycetes were surveyed for GH18-CBM5 sequences, and no hits were found for Ascomycetes (see below). Therefore, GH18 sequences from Ascomycetes were excluded from further analysis. In total, 985 fungal GH18 sequences, 212 of which contained CBM5 motifs, were extracted. From the GH18 domains of the *Si* and *Sv* chitinases, a Hidden Markov model (HMM) profile was built using the hmmbuild function from the hmmer[67] package (version 3.3.2) in the Linux terminal. This GH18 hmm profile was used as input for the hmmsearch function from the same package to search against the 985 downloaded fungal sequences. From the hmmsearch results, sequences with a best domain score higher than 200 were selected. In total, 240 fungal sequences passed this threshold, 51 of which had the CBM5 motif. All sequences belonged to the phylum Basidiomycota. The GH18 domains of these sequences were extracted using the same hmmsearch function with the –domtblout argument, obtaining the starting and ending position of the GH18 HMM domain per sequence from the envelope columns. For sequences that had multiple GH18 domains, the GH18 domain closest to the C-terminal was chosen. Only the GH18 domain of these sequences was used in the subsequent analysis (Supplementary File 1).

### Bacterial sequence extraction

Bacterial GH18 sequences were extracted from UniprotKB(release 2025_04) using the following search terms: (taxonomy_id:2) AND (xref:pfam-PF00704). Taxonomy ID 2 refers to bacteria and PF00704 refers to the Pfam annotation for the GH18 motif. The hmmsearch of the *Si/Sv* GH18 hmm profile was performed against the bacterial sequences, similarly to what was done before the extraction of the GH18 domain. Based on this, two types of outgroup sequences were chosen. The first group contained bacterial sequences that had a best domain score of more than 200. 5 GH18 sequences were picked for this group, corresponding to 3 bacterial species that were reported to live in soil and/or colonized plants. For two of the three species, two

sequences were chosen, one from a CBM5-containing sequence and one without. From the third species, only the CBM5-containing sequence passed the threshold. The second bacterial group contained GH18 sequences with a low best domain score (between 20 and 30) from the hmm search results. 5 random sequences were chosen from that group, and they were from non-CBM5-containing sequences.

### Sequence-based distinction between CBM5 and CBM12

In the InterPro and Pfam annotations, CBM5 and CBM12 domain sequences are not distinguished. To clearly separate the CBM5 motif and the closely related CBM12 motif, CBM5 and CBM12 sequences were downloaded from the CAZy[68] database. Then, sequences found in both entries were discarded to remove any potential overlap. 20 random sequences per the remaining CBM5 or CBM12-specific sequences were chosen to build a respective HMM profile using the hmmbuild from hmmer package (version 3.3.2) (Supplementary Fig. 17). Next, an hmmsearch using either the CBM5 or CBM12 hmm profile was executed against either CBM5 or CBM12-specific sequences that were not included in building the hmm profile. For the CBM5 sequences, the best domain score from the hmm search of both the CBM5 and CBM12 motifs was compared. The same was done for the CBM12 sequences. For CBM12-specific sequences, the CBM12 hmm score was higher than the CBM5 score ($n = 162$; Wilcoxon test; $p$-value $< 2.2\mathrm{e}{-16}$; $V = 13162$). For CBM5- specific sequences, the corresponding CBM5 hmm score was higher ($n = 164$, Wilcoxon test; $p$-value $< 2.2\mathrm{e}{-16}$; $V = 13108$).

To confirm that the CBM5 sequences used were truly CBM5 and not CBM12, the CBM5 and CBM12 hmm search was applied on them, which confirmed that the CBM5 hmm score is higher than the CBM12 one.

### Sequence processing and phylogenetic inference

GH18 sequences were aligned using MAAFT[69] (version 7). After the alignment, the sequences were trimmed to remove uninformative sites using Clipkit[70] (Version 2.6.1) on the Linux terminal. Next, the sequences were used as input for IQ-TREE[71] (version 2.0.7) on the Linux terminal. Model inference without tree reconstruction was conducted using the -MF argument to determine the optimal evolutionary model using 31 threads with the -T 31 argument. The most optimal model was WAG + R10 and was used for tree construction using the argument -m WAG + R10 with 31 threads. Branch support was calculated using Ultrafast bootstrapping with the argument -B 1000.

### Tree manipulation and graphing

All processes related to tree visualization were done using Evolview[72] (version 2).

### Reporting summary

Further information on research design is available in the Nature Portfolio Reporting Summary linked to this article.

## Data availability

The phylogeny data used in this study are available in the Supplementary Data 1 file. Source data are provided with this paper.

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

## Acknowledgements

We acknowledge support from the Cluster of Excellence on Plant Sciences (CEPLAS) funded by the Deutsche Forschungsgemeinschaft (DFG, German Research Foundation) under Germany's Excellence Strategy-EXC 2048/1-Project ID: 390686111 and the ZU 263/11-2 (SPP 2125 DECRyPT).

## Author contributions

R.E. and A.Z. conceptualized the study. R.E. and A.E. conducted most of the research. M.J.H., B.M.M., and R.E. conducted mass spectrometric analysis of the product profile and the subsite specificity. T.N. conducted the phylogenetic analysis of fungal GH18 chitinases. R.E. and A.Z. wrote the manuscript with input from all authors.

## Funding

## Competing interests

The authors declare no competing interests.
