## [Transparent Peer Review file · Nature Communications]

Domain gain or loss in a fungal chitinase enables specialization towards antagonism or immune suppression

Corresponding Author: Professor Alga Zuccaro

Version 0:

Reviewer comments:

Reviewer #1

(Remarks to the Author)

This manuscript addresses an interesting and timely topic, exploring how structural modularity in fungal chitinases may drive ecological specialization between microbial antagonism and immune suppression. The work is interesting and of potential interest to the Nature Communications readership. However, the current version remains largely descriptive and lacks the depth and mechanistic insight needed to fully support some of its central claims. Several conclusions appear overstated given the evidence provided, and important experimental details, controls, and quantitative analyses are missing. Our detailed comments below highlight areas where the authors could strengthen the work through additional experiments, more rigorous data analysis, and deeper integration of their findings within the broader literature.

- Page 4, lines 99-105. Stating that the antagonism is mediated by a GH18-CBM5 is an overstatement and should be rephrased. Although Eichfeld et al 2024 (published by the authors of the current manuscript) reported some evidence of a role for this GH18-CBM5 in inhibiting the pathogen *B. sorokiniana*, many other genes were found to be induced during this interaction, therefore there is no evidence that GH18-CBM5 is THE mediator.
- Page 6, lines 148-150: the authors should also test activity on what they later on call “protein-free *Bs* mycelium” (since they will show binding of the GH18-CBM5 to it, “indicating that the domain contributes to cell wall targeting”, hence the enzyme should also be active on this same substrate). A time course experiment is only provided for chitin azure, but should be done also for crab shell chitin and chitosan.
- Fig 1: should absorbance data from assays (DNSA, chitin azure, protein binding/bradford) better be converted to more relevant values via standard curves? (also for Fig4); would be nice if TLC plate had a chito-oligo standard/ladder for comparison, not just GlcNAc
- Proper kinetics of SiCHIT and SiCHIT2 (both with and without CBM5, and SCHIT silenced mutant) should be determined in order to properly compare the enzymes and their substrate preference.
- Fig 1B and C, and 2 A-C, S2, Fig 4, Fig 5 need some statistics for significance (P value).
- Page 6 and Fig 1E: the part on mass spectrometry feels a bit superficial in the main text, and the corresponding details in Methods seem to be completely missing. What mass spec instrument was used, how were samples prepared/analysed, etc? Why colloidal chitin, and not crab shell chitin or mycelium chitin? The author should also include the original mass spec results in Supp Info to prove the identity of the products, as just Fig 1E is not enough. Same for Fig 1F.
- Conclusion in p6 line 166/167: less binding to chitosan may also be due to molecule size of chitosan compared to chitin and *Bs* cell walls/mycelium, the authors don't specify molecular weight (low, medium, high).
- Fig S1: would be nice to indicate the expected molecular weight of the recombinant proteins
- Page 7, line 165: what is this protein-free *Bs* mycelium and how was it made? Also, p7 line 172 “substrate binding, particular to ...fungal cell wall-like chitin” the authors just show pull down of mycelium, not specifically the chitin in the cell walls....
- Fig 1G. This is fine, but it would be helpful to confirm it by checking bound and unbound protein using SDS-PAGE. Easy experiment to do.
- Page 7: I do not think that treating *Bs* spores with Si chitinase is enough to conclude that the enzyme has an effect on pathogen growth in planta. What about the effect on *Bs* mycelium? The binding experiment was done on mycelium (not spores), after all, plus the mycelium may be more relevant with the actual type of in vivo interaction that takes place between the two fungi in a plant. Ideally such an experiment should be done in planta, but the alternative would be in agar plates where the chitinase is added and the growth of *Bs* mycelium measured. Same should be done on Si itself, to rule out

negative effects of the chitinase on the producing organism.

- There is some lack of consistency in presenting data. Fig 2 and S2 show similar results obtained from barley and Arabidopsis, respectively, yet the graphs are shown as box plots in Fig 2, and histograms in S2.
- Page 8, line 208: please state the % of sequence identity.
- Fig 3 F) explain scale -2 to 3 L2FC; why Hv only 3 and 10 dpi?
- What AlphaFold version was used?
- P9 line 228 missing reference to Fig 3 FG, while no data shown for SiCHIT 3 and 4
- Page 9, line 235-236. This statement is not in line with Fig 3I, which clearly shows that SiCHIT was induced on crab shell chitin, and SiCHIT2 was not.
- Fig 4 B and C: a time course here is really needed, as we might be looking at plateau activity.
- Figure 5 C: Statistical significance MUST be shown here. SiCHIT2 (without linker) should also be assessed.
- Figure 5D is not mentioned at all in the results. Importantly, Fig 5D seems to contradict the ROS experiment. SiCHIT does not repress ROS, but does not trigger defense gene activation (over EV controls)? And SiCHIT-cbm prevents ROS generation but induces highest levels of defense gene expression? While SiCHIT2 represses ROS but induces similar levels of defense gene in barley as SiCHIT2+CBM....
- Fig 6 not mentioned in manuscript? Or typo p11 line 302 ("Fig 5")?
- P11 line 281 Can you call it immune-suppressive function or is it rather the evasion of immune reaction?
- The authors should check what happens to chitohexaose upon incubation with SiCHIT+CBM5, SiCHIT-CBM5, SiCHIT2-CBM5 and SiCHIT2+CBM5 (a time course would be best). What products are generated? Probably, the presence of CBM5 has some effect on the mechanism of action of the GH. The authors themselves state "These findings suggest that CBM5 may hinder activity on soluble substrates not by altering catalytic specificity, but by limiting access to short oligomers through spatial constraints", however they have not provided experimental evidence. It would also be valuable to investigate the binding affinity of the enzymes for chitohexaose compared to polymeric chitin substrates. Such analyses would help elucidate the mechanistic basis for the functional differences between enzymes with or without CBM5 and their roles as effectors.
- In addition, the authors should address how the fungal chitinase can attack the cell wall of *Bipolaris* while leaving the producing fungus's own cell wall intact. This is a key, yet lacking bit of information.
- Reference required for p3 line 66-68? P4 line 88?, (p4 line 91 own work cited), p4 line 94? P4 line 98; General: 10 out of 37 ref in total are from Zuccaro lab...

Reviewer #2

(Remarks to the Author)

This manuscript presents a compelling investigation into the functional divergence of GH18 chitinases in the mutualistic fungus *Serendipita indica*, with particular emphasis on how gain or loss of a CBM5 domain determines ecological specialization between microbial antagonism and host immune suppression. The study is conceptually interesting and addresses an important topic in fungal biology. The combination of biochemical assays, domain-swapping experiments, and host-pathogen interaction studies offers potentially valuable insights into effector modularity of symbiosis-related functions.

My major concerns are outlined below:

1. While the study demonstrates that CBM5 is required for the antifungal function of SiCHIT, the mechanistic basis of CBM5's contribution remains superficially explored. It is unclear whether this function is unique to CBM5 or shared among other CBM family members. I suggest the authors discuss or test whether other CBM families could confer similar antifungal properties when fused to SiCHIT or SiCHIT2. This would help determine whether the observed effect is CBM5-specific or reflects a broader property of chitin-binding domains.
2. The manuscript focuses primarily on two paralogs, SiCHIT and SiCHIT2, but it is important to understand the functional diversity of the complete GH18 chitinase repertoire in *Serendipita indica*. The authors mention four GH18 chitinases in the genome but provide limited functional or expression analysis beyond the two focal genes. I recommend a more systematic analysis of all GH18 family members, including their domain architecture, expression patterns under relevant conditions, and potential functional roles.
3. A significant methodological limitation is the inconsistent inclusion of SiCHIT2 as a control across functional assays. Since SiCHIT2 is a closely related paralog lacking CBM5, it should be included alongside SiCHIT and SiCHIT-CBM5 in all comparative experiments, particularly in antifungal and immune suppression assays. Including SiCHIT2 throughout would provide a clearer baseline for interpreting how CBM5 alters function and would strengthen the conclusion regarding modularity and functional divergence.
4. The conclusion that immune-suppressive effectors evolved from antimicrobial ancestors primarily based on the presence or absence of a CBM5 domain in a single species is an oversimplification. Domain gain or loss is only one component of effector evolution, and without a broader comparative analysis across multiple species and fungal lineages, the evolutionary trajectory remains speculative. I recommend that the authors either expand their phylogenetic and comparative domain analysis or moderate their claims regarding the evolutionary origin of chitinase modularity.

Reviewer #3

(Remarks to the Author)

This study, entitled "Domain gain or loss in fungal chitinases drives ecological specialization toward antagonism or immune

suppression,” is an elegant study of how fungal effector proteins evolve through structural modularity and gene duplication. Focusing on two paralogous chitinases found in tandem — SiCHIT and SiCHIT2 — the authors dissect how gain or loss of the C-terminal carbohydrate-binding module CBM5 drives divergent ecological roles, including fungal antagonism and host immune suppression.

The fusion and deletion experiments demonstrate the modular nature of CBM5. Its presence equips chitinases for pathogen defense, while its absence tunes the enzyme for immune evasion. By tracking expression patterns, domain architecture, and protein functionality, the authors present a persuasive case for evolutionary repurposing via gene duplication and domain loss, hypothesizing a transition from an antimicrobial effector role to an effector involved in immune suppression. The work beautifully ties molecular findings to ecological specialization, showing how domain structure contributes to niche adaptation in symbiotic fungi.

More broadly, this study provides significant mechanistic insight into effector biology by demonstrating how structural modularity enables functional diversification. Given its potential relevance across a wide range of species, these findings have far-reaching implications. The depth and breadth of the work make it well-suited for publication in Nature Communications.

I only have a few minor comments/suggestions

-One of the things I was struck by was the transcriptional profile differences (regulatory rewiring) that were discovered, which are likely critical to the functional adaptation. Did the authors look at the promoter region of the paralogs to see if there was a simple explanation for this? I don't know if this is necessary for the paper, but it would be interesting to know if the duplication included the same promoter region as part of the adaptive evolution.

-This point may be nitpicky, but in both the methods (line 414-415) and the results (line 248), the authors mention the suppressed germination. I would argue that germination is not suppressed, but rather, the chitinase is degrading the cell wall of the new hyphae to inhibit growth rather than actually suppressing germination.

Line 196 -Should be "...required for full SiCHIT's antifungal..."

Figure 5A – this is difficult to follow. The authors may want to consider a different color scheme for the different treatments.

Overall, this is an excellent story, and it was enjoyable to review.

Reviewer #4

(Remarks to the Author)

Version 1:

Reviewer comments:

Reviewer #1

(Remarks to the Author)

The authors have satisfactorily addressed most of my concerns and significantly strengthened the manuscript by including additional data, however there are still some points that need clarification and additional evidence:

- Point 10. I still cannot find the methods for how protein-free Bs mycelium was made. It actually seems that what was called “protein-free Bs mycelium” in the first manuscript has now been changed to “washed Bs mycelium” in the revised version.

- Point 11. Just for clarity, Fig 1G is now 1H. Importantly, how can up to 17.5uM protein remain unbound, when the input only was 4uM (as stated in Supp Fig 5 and in methods)? Where are the input only values in Fig 1H?

- Point 24. The DNS assay is useful, however it shows that SiChit2 has only about 30% more activity on chitohexaose than SiChi, yet the difference in ROS signal is massive. This must be due to the type of products generated from cleavage of the substrate. The authors should definitely check the profile of these products (and their abundance) and how they change over time upon incubation with SiChit, SiChitE196Q, SiChit-CBM5, SiChit2+link and SiChit2+CBM5, using e.g. HPAEC or alternative approaches.

Reviewer #2

(Remarks to the Author)

Thank you for the authors' careful and constructive responses.

Reviewer #3

(Remarks to the Author)

The authors have adequately addressed the reviewers concerns.

Reviewer #4

(Remarks to the Author)

Point-by-point response

We thank the Editor and all four reviewers for their careful evaluation of our manuscript and for their constructive and insightful feedback. We have addressed all comments through additional experiments, expanded analyses, and substantial revision of the text. Below, we provide a detailed, point-by-point response to each comment.

Response to reviewers

Reviewer # 1/4: This manuscript addresses an interesting and timely topic, exploring how structural modularity in fungal chitinases may drive ecological specialization between microbial antagonism and immune suppression. The work is interesting and of potential interest to the Nature Communications readership. However, the current version remains largely descriptive and lacks the depth and mechanistic insight needed to fully support some of its central claims. Several conclusions appear overstated given the evidence provided, and important experimental details, controls, and quantitative analyses are missing. Our detailed comments below highlight areas where the authors could strengthen the work through additional experiments, more rigorous data analysis, and deeper integration of their findings within the broader literature.

1. Page 4, lines 99-105. Stating that the antagonism is mediated by a GH18-CBM5 is an overstatement and should be rephrased. Although Eichfeld et al 2024 (published by the authors of the current manuscript) reported some evidence of a role for this GH18-CBM5 in inhibiting the pathogen *B. sorokiniana*, many other genes were found to be induced during this interaction, therefore there is no evidence that GH18-CBM5 is THE mediator. **Reply:** We agree that the previous phrasing overstated the role of GH18-CBM5. We have rephrased the section to clarify that SiCHIT is not the sole mediator of antagonism, but rather strongly contributes to the overall antifungal activity of *S. indica*. The revised text now states: "This antagonism involves multiple molecular factors, with a GH18-CBM5 endochitinase (SiCHIT) playing a prominent role by restricting *Bs* growth and recapitulating the protective effect of *Si in planta*."
2. Page 6, lines 148-150: the authors should also test activity on what they later on call protein-free *Bs* mycelium (since they will show binding of the GH18-CBM5 to it, indicating that the domain contributes to cell wall targeting, hence the enzyme should also be active on this same substrate). **Reply:** We agree that assessing enzyme activity on protein-free *B. sorokiniana* mycelium is important, given that SiCHIT binds this substrate. We performed the additional assay and confirmed that the enzyme is active on protein-free fungal cell wall material. These results are now included in the revised manuscript (Supplementary Fig. 14) and referenced in the Results section.
3. A time course experiment is only provided for chitin azure, but should be done also for crab shell chitin and chitosan. **Reply:** We performed additional time-course assays using both crab shell chitin and chitosan, analogous to the chitin-azure experiment. The new data are included in the revised manuscript (Fig. 1) and are described in the Results section.
4. Fig 1: should absorbance data from assays (DNSA, chitin azure, protein binding/bradford) better be converted to more relevant values via standard curves? (also for Fig4); would be nice if TLC plate had a chito-oligo standard/ladder for

comparison, not just GlcNAc. **Reply:** We agree that converting absorbance measurements to quantitative values is preferable where possible. Accordingly, we recalculated the DNSA reducing-end assays for crab shell chitin and chitosan, as well as the protein–substrate binding assays, using appropriate standard curves. For the chitin-azure assay, such a conversion is not feasible because the substrate consists of chitin covalently linked to Remazol Brilliant Violet 5R (RBV); the soluble RBV–chitin fragments released during hydrolysis do not allow the generation of a defined standard curve. In response to the request for a more informative product analysis, we removed the TLC panel and replaced it with a mass spectrometric characterization of the reaction products generated from crab shell chitin. This provides substantially higher resolution and more detailed information than the original TLC assay.

- 5. Proper kinetics of SiCHIT and SiCHIT2 (both with and without CBM5, and SCHIT silenced mutant) should be determined in order to properly compare the enzymes and their substrate preference.** **Reply:** We agree that kinetic parameters are important for comparing substrate preferences of the different chitinases. We therefore performed Michaelis–Menten analyses for all relevant constructs, including SiCHIT, SiCHIT lacking the CBM5, SiCHIT2, and SiCHIT2 fused to CBM5. The results are provided in the revised manuscript (Supplementary Fig. 13).

These analyses show that the presence of the CBM5 reduces the apparent K_m , consistent with enhanced substrate binding, but slightly decreases V_{max} , suggesting a modest constraint on maximal catalytic turnover under substrate-saturating conditions. We want to mention here that these parameters should be interpreted cautiously because classical Michaelis–Menten assumptions do not fully apply to reactions involving insoluble polysaccharide substrates. Nevertheless, the data allow for robust comparative assessments and provide additional mechanistic insight into how CBM5 modulates enzyme performance.

- 6. Fig 1B and C, and 2 A-C, S2, Fig 4, Fig 5 need some statistics for significance (P value).** **Reply:** Each figure legend includes a dedicated section describing the statistical tests performed, the number of biological replicates, and the multiple-testing correction applied. For multi-factorial comparisons, we indicate statistically significant differences (adjusted p-value < 0.05) using a letter-code grouping scheme, as is standard practice. This ensures consistent and transparent presentation of statistical significance across all datasets.
- 7. Page 6 and Fig 1E: the part on mass spectrometry feels a bit superficial in the main text, and the corresponding details in Methods seem to be completely missing. What mass spec instrument was used, how were samples prepared/analysed, etc? Why colloidal chitin, and not crab shell chitin or mycelium chitin? The author should also include the original mass spec results in Supp Info to prove the identity of the products, as just Fig 1E is not enough. Same for Fig 1F.** **Reply:** we replaced the colloidal chitin sample with a mass-spectrometric product profile generated from crab shell chitin in Fig. 1, which represents a more relevant substrate for our study. We also added the catalytically impaired SiCHIT^{E196Q} variant as a control to illustrate the expected residual activity. In addition, we included a product profile for β -chitin in the Supplementary Information (Supplementary Fig. 3). To address the reviewer's request for methodological clarity, we substantially expanded the Methods section with full details on sample

preparation, instrumental parameters, and data analysis. We also now provide representative original mass spectra in Supplementary Figs. 2 and 4.

8. **Conclusion in p6 line 166/167: less binding to chitosan may also be due to molecule size of chitosan compared to chitin and BS cell walls/mycelium, the authors should specify molecular weight (low, medium, high).** **Reply:** We thank the reviewer for this thoughtful point. The average molecular weight of the chitosan used in our assays (346 kDa) has now been added to the Materials and Methods section. We agree that polymer size and surface characteristics can influence absolute binding efficiencies. However, in our analysis we compare the binding behavior of the two enzymes on the same substrate (crab shell chitin, >75% de-acetylated chitosan, or protein-free *B. sorokiniana* mycelium). Because each comparison is made within a single substrate type, differences in molecular weight do not confound our conclusions regarding the relative binding properties of the enzymes.
9. **Fig S1: would be nice to indicate the expected molecular weight of the recombinant proteins.** **Reply:** We have added the expected molecular weights of all recombinant proteins to the legend of Supplementary Fig. S1 to facilitate interpretation.
10. **Page 7, line 165: what is this protein-free Bs mycelium and how was it made? Also, p7 line 172 substrate binding, particular to fungal cell wall-like chitin the authors just show pull down of mycelium, not specifically the chitin in the cell walls.** **Reply:** We have now added a description of how protein-free *B. sorokiniana* mycelium was generated to the Materials and Methods section. We also revised the corresponding section in the Results to clarify that the binding assays were performed on protein-depleted fungal cell wall material enriched in chitin rather than on isolated chitin. This rephrasing more accurately reflects the nature of the substrate and the interpretation of the binding experiments.
11. **Fig 1G. This is fine, but it would be helpful to confirm it by checking bound and unbound protein using SDS-PAGE. Easy experiment to do.** **Reply:** We agree that confirming the binding assay with an SDS-PAGE analysis strengthens the result. We therefore analyzed bound and unbound fractions for the relevant enzyme-substrate combinations by SDS-PAGE. The results, which fully support the binding patterns shown in Fig. 1G, are now included in the Supplementary Information (Supplementary Fig. 5).
12. **Page 7: I do not think that treating Bs spores with Si chitinase is enough to conclude that the enzyme has an effect on pathogen growth in planta. What about the effect on Bs mycelium? The binding experiment was done on mycelium (not spores), after all, plus the mycelium may be more relevant with the actual type of in vivo interaction that takes place between the two fungi in a plant. Ideally such an experiment should be done in planta, but the alternative would be in agar plates where the chitinase is added and the growth of Bs mycelium measured. Same should be done on Si itself, to rule out negative effects of the chitinase on the producing organism.** **Reply:** We appreciate the reviewer's concern and agree that both spore- and mycelium-directed activities can contribute to antifungal outcomes in planta. To clarify this point, we would like to emphasize that *S. indica* does not interact exclusively with *B. sorokiniana* hyphae during colonization. Rather, our recent imaging data show that *Si* hyphae establish direct physical contact with *Bs* spores on barley roots, indicating that spores represent an ecologically relevant target during the early stages of the interaction. To make this explicit, we show representative confocal

images below that illustrate key steps of the Si–Bs interaction in planta: (i) Si hyphae coil around Bs spores, (ii) translocate antimicrobial effector proteins into the spores (e.g. Dld1), and (iii) penetrate Bs spores. These observations demonstrate that Si engages in a mycoparasitic attack on Bs at the spore stage, confirming that antifungal activity directed toward spores is biologically meaningful and relevant for early pathogen suppression. We agree that a more extensive mechanistic analysis of Si-mediated antagonism, including the roles of additional effector proteins, would be valuable. A comprehensive dataset addressing these aspects is part of a separate manuscript currently in preparation, and we therefore prefer not to include it here to avoid fragmenting that study. Nevertheless, we hope that the microscopy data provided below clearly show that spore-targeted antifungal activity is not an artificial assay choice but reflects a genuine interaction interface in planta. Finally, we note that potential negative effects of SiCHIT on the producing organism have already been assessed in our previous work (Eichfeld et al. 2024), where treatment with the enzyme did not impair *S. indica* growth.

a) **Features of mycoparasitism in *S. indica***

Features of mycoparasitism of *Si* confronted with *Bs* during established host colonization. Arabidopsis Col-0 roots were colonized with *Si* to establish a mutualistic interaction and subsequently challenged with *Bs*. *Si* cell wall structures were stained with the wheat germ agglutinin Alexa Fluor 488 conjugate (488-WGA in green).

13. **There is some lack of consistency in presenting data. Fig 2 and S2 show similar results obtained from barley and Arabidopsis, respectively, yet the graphs are shown a box plots in Fig 2, and histograms in S2.** **Reply:** We have now harmonized the data presentation accordingly.
14. **Page 8, line 208: please state the % of sequence identity.** **Reply:** We have now added the corresponding percentage sequence identity to the text at the indicated location.
15. **Fig3 F) explain scale -2 to 3 L2FC; why Hv only 3 and 10 dpi?** **Reply:** We have clarified both points in the revised figure legend for Fig. 3. The legend now explains the log₂ fold-change (-2 to 3) scale used in panel F, and it provides the rationale for including barley (Hv) expression data specifically at 3 and 10 dpi, which represent the biologically most informative stages for comparing SiCHIT family gene expression during barley colonization.
16. **What AlphaFold version was used?** **Reply:** We have included that information in the figure legend of Figure 3.
17. **P9 line 228 missing reference to Fig 3 FG, while no data shown for SiCHIT 3 and 4.** **Reply:** We have now added the appropriate reference to Fig. 3F–G in the text. In addition, we have included new expression data for SiCHIT3 and SiCHIT4 in the revised manuscript, allowing direct comparison of all four GH18 chitinases under the relevant conditions.
18. **Page 9, line 235-236. This statement is not in line with Fig 3I, which clearly shows that SiCHIT was induced on crab shell chitin, and SiCHIT2 was not.** **Reply:** We thank the reviewer for catching this inconsistency. We have revised the text at the indicated location to accurately reflect the expression patterns shown in Fig. 3I.
19. **Fig 4 B and C: a time course here is really needed, as we might be looking at plateau activity.** **Reply:** We performed the corresponding time-course experiments and have included the results in the Supplementary Information (Supplementary Fig. 12). The relevant section of the Results has been updated accordingly to incorporate and interpret these new data.
20. **Figure 5 C: Statistical significance MUST be shown here. SiCHIT2 (without linker) should also be assessed.** **Reply:** We have repeated the experiment under the same conditions and incorporated the new data into our analysis to strengthen the statistical power. Regarding the suggestion to assess SiCHIT2 without the linker region, we do not consider this construct informative for our study. The native SiCHIT2 protein contains a short C-terminal segment that does not connect to any additional domain and likely represents a residual linker structure retained after domain loss (see Fig. 3 and related text). Because this region is part of the natural architecture of SiCHIT2, removing it would also generate an artificial construct that does not reflect the endogenous enzyme. For this reason, and because the linker does not have a functional role comparable to the CBM5 domain in SiCHIT, we did not pursue a SiCHIT2 construct lacking this region.
21. **Figure 5D is not mentioned at all in the results. Importantly, Fig 5D seems to contradict the ROS experiment. SiCHIT does not repress ROS, but does not trigger**

defense gene activation (over EV controls)? And SiCHIT-cbm prevents ROS generation but induces highest levels of defense gene expression? While SiCHIT2 represses ROS but induces similar levels of defense gene in barley as SiCHIT2+CBM. **Reply:** We have now incorporated explicit references to Fig. 5D in the Results section and substantially revised the accompanying text to clarify the interpretation of these data. In response to the reviewer's concern, we also added new experimental data to Fig. 5 to strengthen the conclusions drawn from both the ROS and defense gene—expression assays. The perceived discrepancy between ROS suppression and PR10 induction largely reflects differences in the timing and nature of the readouts. The transient ROS burst occurs within minutes after chitin perception, whereas PR10 transcript accumulation is measured at later time points and reflects the overall interaction outcome rather than the immediate immune response. The slight (non-significant) increase in PR10 expression observed for SiCHIT-CBM5 and SiCHIT2+link is most likely explained by enhanced Si colonization at this stage, rather than by a failure to evade the early immune response. In other words, the immune-evasive activity of the recombinant chitinases appears to provide a short-term advantage that facilitates initial colonization (“kick-start”), but does not persist as a long-lasting suppression of all downstream immune outputs in our experimental setup. We hope that the additional data and revised explanation address the reviewer's concern.

22. **Fig 6 not mentioned in manuscript? Or typo p11 line 302 (1C;Fig 5D;)?** **Reply:** Thank you for noting this. It was a typographical error, and we have corrected it in the revised manuscript.
23. **P11 line 281 Can you call it immune-suppressive function or is it rather the evasion of immune reaction?** **Reply:** We agree that “immune suppression” implies a different mechanism. We have therefore rephrased the text.
24. **The authors should check what happens to chitohexaose upon incubation with SiCHIT+CBM5, SiCHIT-CBM5, SiCHIT2-CBM5 and SiCHIT2+CBM5 (a time course would be best). What products are generated? Probably, the presence of CBM5 has some effect on the mechanism of action of the GH. The authors themselves state. These findings suggest that CBM5 may hinder activity on soluble substrates not by altering catalytic specificity, but by limiting access to short oligomers through spatial constraints, however they have not provided experimental evidence. It would also be valuable to investigate the binding affinity of the enzymes for chitohexaose compared to polymeric chitin substrates. Such analyses would help elucidate the mechanistic basis for the functional differences between enzymes with or without CBM5 and their roles as effectors.** **Reply:** We have tested the release of reducing ends from chitohexaose at different time points and included the data in the supplementary part of the manuscript (Supplementary Figure 16). Our results are in line with findings on bacterial GH18 and GH19 chitinases where the CBM5 enhances degradation of insoluble substrates but interferes with degradation of small, soluble substrates^{1,2}. We have also discussed this in the current version (p 13, line 366 – 372). It also aligns well with our kinetic analysis (Supplementary Figure 13) that demonstrates that the CBM5 can slightly interfere with maximal reaction rates under substrate saturating conditions, highlighting the functional trade-off that comes with presence of the CBM5.

- 25. In addition, the authors should address how the fungal chitinase can attack the cell wall of *Bipolaris* while leaving the producing fungus own cell wall intact. This is a key, yet lacking bit of information. Reply:** We agree that this is an important and fascinating question. Similar issues have been discussed for other mycoparasitic and antagonistic fungi³, where several mechanisms have been proposed, including cell wall modifications, secretion of protective lectins, or highly localized expression of chitinases. A full experimental investigation of how *S. indica* protects its own cell wall while targeting *B. sorokiniana* would, however, constitute a substantial study on its own and is beyond the scope of the current manuscript. We have now added a brief discussion of these possible mechanisms to the revised manuscript (p. 14, lines 394–401) to acknowledge this important aspect and to highlight it as a direction for future research.
- 26. Reference required for p3 line 66-68? P4 line 88?, (p4 line 91 own work cited), p4 line 94? P4 line 98; General: 10 out of 37 ref in total are from Zuccaro lab. Reply:** We have added the appropriate references at all indicated positions and incorporated additional citations throughout the introduction and discussion to better contextualize our work within the broader literature. While some self-citation is unavoidable given our previous studies on *S. indica* effectors and inter-microbial interactions, we have reduced the overall proportion of our own references and ensured that all claims are now supported by external sources where available.

Reviewer #2: This manuscript presents a compelling investigation into the functional divergence of GH18 chitinases in the mutualistic fungus *Serendipita indica*, with particular emphasis on how gain or loss of a CBM5 domain determines ecological specialization between microbial antagonism and host immune suppression. The study is conceptually interesting and addresses an important topic in fungal biology. The combination of biochemical assays, domain-swapping experiments, and host-pathogen interaction studies offers potentially valuable insights into effector modularity of symbiosis-related functions. My major concerns are outlined below:

- 1. While the study demonstrates that CBM5 is required for the antifungal function of SiCHIT, the mechanistic basis of CBM5's contribution remains superficially explored. It is unclear whether this function is unique to CBM5 or shared among other CBM family members. I suggest the authors discuss or test whether other CBM families could confer similar antifungal properties when fused to SiCHIT or SiCHIT2. This would help determine whether the observed effect is CBM5-specific or reflects a broader property of chitin-binding domains. Reply:** We thank the reviewer for this valuable suggestion. We have now added a dedicated section in the Discussion (p. 12, line 330 ff.) addressing the question of CBM5 specificity. In line with the reviewer's point, we note that while CBM5 is the canonical chitin-binding module associated with GH18 chitinases in Basidiomycota, other CBM families with comparable chitin-binding architectures or residue compositions could, in principle, substitute for CBM5 and support similar antifungal activities. The extent to which this is possible likely depends on the precise binding mode and affinity contributed by each CBM type. Experimental testing of alternative CBMs is an interesting direction for future work and is now explicitly mentioned in the manuscript.

2. The manuscript focuses primarily on two paralogs, SiCHIT and SiCHIT2, but it is important to understand the functional diversity of the complete GH18 chitinase repertoire in *Serendipita indica*. The authors mention four GH18 chitinases in the genome but provide limited functional or expression analysis beyond the two focal genes. I recommend a more systematic analysis of all GH18 family members, including their domain architecture, expression patterns under relevant conditions, and potential functional roles. **Reply:** We appreciate this suggestion and have expanded our analysis accordingly. We now provide additional information on all four GH18 chitinases in the Results section, including their domain architectures and structural predictions. Structural models for SiCHIT3 and SiCHIT4 have been added to the Supplementary Information (Supplementary Fig. 10). Furthermore, we have integrated expression data for SiCHIT3 and SiCHIT4 under the relevant conditions into Fig. 3 to allow direct comparison with SiCHIT and SiCHIT2. Finally, we have expanded the Discussion (p. 14, lines 377–393) to address the potential functional roles of each GH18 family member within the broader context of *S. indica* biology.
3. A significant methodological limitation is the inconsistent inclusion of SiCHIT2 as a control across functional assays. Since SiCHIT2 is a closely related paralog lacking CBM5, it should be included alongside SiCHIT and SiCHIT-CBM5 in all comparative experiments, particularly in antifungal and immune suppression assays. Including SiCHIT2 throughout would provide a clearer baseline for interpreting how CBM5 alters function and would strengthen the conclusion regarding modularity and functional divergence. **Reply:** We thank the reviewer for this comment. SiCHIT2 (SiCHIT2^{+link}) is included in all comparative experiments from Fig. 4 onward, including the biochemical assays, antifungal assays, and immune-suppression assays. These datasets allow direct comparisons between SiCHIT, SiCHIT^{-CBM5}, and SiCHIT2 variants and support our conclusions regarding the functional contribution of the CBM5 domain.
4. The conclusion that immune-suppressive effectors evolved from antimicrobial ancestors primarily based on the presence or absence of a CBM5 domain in a single species is an oversimplification. Domain gain or loss is only one component of effector evolution, and without a broader comparative analysis across multiple species and fungal lineages, the evolutionary trajectory remains speculative. I recommend that the authors either expand their phylogenetic and comparative domain analysis or moderate their claims regarding the evolutionary origin of chitinase modularity. **Reply:** Our expanded analysis shows that GH18 chitinases with and without CBM5 occur intermingled within a single fungal clade. Although our dataset does not allow a definitive reconstruction of ancestral states, the observed distribution suggests that, in many cases, loss of CBM5 is a more parsimonious explanation than multiple independent gains within this clade, given the widespread presence of CBM5-containing variants and the scattered occurrence of GH18 catalytic domains lacking CBM5. Consistent with this, the outgroup of this clade consists of bacterial GH18 chitinases that carry CBM5. We have adjusted the corresponding statements in the manuscript to reflect this more cautious interpretation and now explicitly note that the precise evolutionary history of these modular arrangements remains unresolved.

Reviewer #3: This study, entitled Domain gain or loss in fungal chitinases drives ecological specialization toward antagonism or immune suppression is an elegant study of how fungal effector proteins evolve through structural modularity and gene duplication. Focusing on two paralogous chitinases found in tandem SiCHIT and SiCHIT2 the authors dissect how gain or loss of the C-terminal carbohydrate-binding module CBM5 drives divergent ecological roles, including fungal antagonism and host immune suppression. The fusion and deletion experiments demonstrate the modular nature of CBM5. Its presence equips chitinases for pathogen defense, while its absence tunes the enzyme for immune evasion. By tracking expression patterns, domain architecture, and protein functionality, the authors present a persuasive case for evolutionary repurposing via gene duplication and domain loss, hypothesizing a transition from an antimicrobial effector role to an effector involved in immune suppression. The work beautifully ties molecular findings to ecological specialization, showing how domain structure contributes to niche adaptation in symbiotic fungi.

More broadly, this study provides significant mechanistic insight into effector biology by demonstrating how structural modularity enables functional diversification. Given its potential relevance across a wide range of species, these findings have far-reaching implications. The depth and breadth of the work make it well-suited for publication in Nature Communications.

1. One of the things I was struck by was the transcriptional profile differences (regulatory rewiring) that were discovered, which are likely critical to the functional adaptation. Did the authors look at the promoter region of the paralogs to see if there was a simple explanation for this? I don't know if this is necessary for the paper, but it would be interesting to know if the duplication included the same promoter region as part of the adaptive evolution. **Reply:** We agree that this is an important point. To address it, we analyzed the putative promoter regions (up to 500 bp upstream of the start codon) of all four GH18 chitinase genes and searched for conserved cis-regulatory motifs. We compared these regions to the promoter architecture of an expanded effector gene family in *S. indica* encoding DELD proteins⁴, which served as a reference. Whereas the upstream regions of the DELD family are relatively conserved in their motif composition, our analysis revealed clear differentiation among the promoter regions of the four chitinase genes, consistent with the transcriptional diversification observed in our expression data. We have added these results to the Supplementary Information (Supplementary Fig. 11).
2. This point may be nitpicky, but in both the methods (line 414-415) and the results (line 248), the authors mention the suppressed germination. I would argue that germination is not suppressed, but rather, the chitinase is degrading the cell wall of the new hyphae to inhibit growth rather than actually suppressing germination. **Reply:** We thank the reviewer for this clarification. We have rephrased the relevant sections in both the Methods and Results to state that the chitinase interferes with early hyphal outgrowth by degrading the emerging cell wall, rather than suppressing germination per se.
3. Line 196 -Should be "...required for full SiCHIT's antifungal... **Reply:** We thank the reviewer for noting this. We have corrected the phrasing.

4. **Figure 5A – this is difficult to follow. The authors may want to consider a different color scheme for the different treatments.** **Reply:** We agree with the reviewer that the original color scheme made the panel difficult to interpret. We have updated the colors for the different treatments in Fig. 5A, and the improved contrast should make the differences much clearer.

- 1 Itoh, Y. *et al.* Importance of Trp59 and Trp60 in chitin-binding, hydrolytic, and antifungal activities of *Streptomyces griseus* chitinase C. *Appl Microbiol Biotechnol* **72**, 1176-1184, doi:10.1007/s00253-006-0405-7 (2006).
- 2 Uni, F., Lee, S., Yatsunami, R., Fukui, T. & Nakamura, S. Mutational analysis of a CBM family 5 chitin-binding domain of an alkaline chitinase from *Bacillus* sp. J813. *Bioscience, biotechnology, and biochemistry* **76**, 530-535, doi:10.1271/bbb.110835 (2012).
- 3 Gruber, S. & Seidl-Seiboth, V. Self versus non-self: fungal cell wall degradation in *Trichoderma*. *Microbiology (Reading, England)* **158**, 26-34, doi:10.1099/mic.0.052613-0 (2012).
- 4 Nostadt, R. *et al.* A secreted fungal histidine- and alanine-rich protein regulates metal ion homeostasis and oxidative stress. *New Phytologist* **227**, 1174-1188, doi:https://doi.org/10.1111/nph.16606 (2020).

Response to editor and reviewers

We thank the Editor and the reviewers once again for their thoughtful evaluation and constructive feedback, which has substantially strengthened our manuscript. We are pleased that the reviewers and editor agree we have adequately addressed previous concerns. Below we provide point-by-point responses to the remaining comments.

Point-by-point response

Reviewer #1/4

The authors have satisfactorily addressed most of my concerns and significantly strengthened the manuscript by including additional data, however there are still some points that need clarification and additional evidence:

10. I still cannot find the methods for how protein-free Bs mycelium was made. It actually seems that what was called “protein-free Bs mycelium” in the first manuscript has now been changed to “washed Bs mycelium” in the revised version.

Answer: We thank the reviewer for highlighting this point. The description of the preparation procedure can be found in the Methods section (lines 557–559). To avoid ambiguity, we now consistently use the term “washed Bs mycelium”, which more accurately reflects the experimental procedure used.

11. Just for clarity, Fig 1G is now 1H. Importantly, how can up to 17.5uM protein remain unbound, when the input only was 4uM (as stated in Supp Fig 5 and in methods)? Where are the input only values in Fig 1H?

Answer: We thank the reviewer for identifying this scaling error. We have corrected Figure 1H scale, updated the raw data file to include input-only values, and revised the figure accordingly.

24. The DNS assay is useful, however it shows that SiChit2 has only about 30% more activity on chitohexaose than SiChi, yet the difference in ROS signal is massive. This must be due to the type of products generated from cleavage of the substrate. The authors should definitely check the profile of these products (and their abundance) and how they

change over time upon incubation with SiChit, SiChitE196Q, SiChit-CBM5, SiChit2+link and SiChit2+CBM5, using e.g. HPAEC or alternative approaches.

Answer: We thank the reviewer for this insightful point. We agree that the DNS assay provides only an estimate of total reducing ends and therefore cannot resolve differences in product composition or reaction kinetics that may be critical for immune activation.

To address this, we performed a time-resolved analysis of chitohexaose (A6) degradation by all indicated chitinase variants (SiCHIT, SiCHIT^{E196Q}, SiCHIT–CBM5, SiCHIT2+link, and SiCHIT2+CBM5) using live SEC–RI–MS. This enabled monitoring of both identity and relative abundance of oligomeric products (A2–A6) over time.

Across all active enzymes, we observed qualitatively similar product spectra, with A2 and A3 as the dominant end products and no evidence for distinct or unusual cleavage products. However, CBM5-lacking variants (SiCHIT–CBM5, SiCHIT2+link) depleted immunogenic A6 and intermediate A4 substantially faster than CBM5-containing counterparts.

Because chitin-triggered ROS signaling is both threshold- and time-dependent, this kinetic bias provides a parsimonious explanation for the observed ROS attenuation through accelerated depletion of immunogenic A6/A4 and accumulation of non-eliciting A2/A3 (Liu et al., 2012; Bozsoki et al., 2017; Zang et al., 2024). We note that additional factors in planta (e.g., protein stability/availability and solubility in the apoplast) could further modulate the magnitude of the response.

These new data are provided as Supplementary Fig. 16C and are referenced in the revised Results section.

Fig. 1. Product profiles of *S. indica* chitinases during chitohexaose (A6) degradation. (A) Time-resolved mass spectrometric analysis of A6 incubated with different recombinant chitinase variants, monitored by live SEC-RI-MS at 37 °C. Stacked bars represent relative MS signal intensities of oligomeric products (A2–A6) over time. (B) Average number of cleavage events per A6 molecule calculated from product distributions shown in (A). (C) Relative enzymatic activity of SiCHIT and SiCHIT2 variants with or without CBM5, normalized to the corresponding full-length enzyme. Percentages were calculated from the data shown in (B).

Reviewer #2

My questions have been largely addressed. However, I would like to raise one remaining concern regarding the interpretation of the CBM5-associated phenotype. The current conclusion appears overstated. From a biochemical perspective, the primary function of the CBM5 domain is to enhance substrate binding and facilitate the proximity of the catalytic GH18 domain to chitin. While the resulting phenotypic consequences may manifest as immune suppression or microbial antagonism, these effects are likely secondary outcomes of improved substrate engagement rather than evidence that CBM5 has evolved a fundamentally new immune-related function. Therefore, I suggest the authors moderate their claims about functional “gain” or “loss”, in evolution and clarify that the core activity remains chitin hydrolysis, with CBM5 mainly modulating efficiency and context of substrate targeting.

Answer: We thank the reviewer for this thoughtful comment and agree that CBM5 primarily functions as a carbohydrate-binding module that enhances substrate engagement for the GH18 catalytic domain, rather than conferring a new catalytic activity. At the same time, our data show that the presence of CBM5 strongly enhances antifungal activity and plant protection by SiCHIT and is sufficient to confer these antagonistic functions when fused to the paralog SiCHIT2. Conversely, CBM5-lacking variants more efficiently deplete soluble immunogenic chitooligomers, suppress chitin-triggered ROS, and promote early root colonization. Thus, CBM5 does not change product specificity but shifts substrate targeting and reaction dynamics in a context-dependent manner, leading to distinct outcomes in microbe–microbe and host–microbe interactions. We have moderated our wording throughout the manuscript, including the title, and explicitly emphasize that the core activity remains chitin hydrolysis. We added the following clarification to the Discussion: “Biochemically, our data support the view that CBM5 primarily enhances substrate engagement and positioning of the GH18 catalytic domain on crystalline chitin. Its presence or absence is sufficient to shift the efficiency and context

of chitin hydrolysis in ways that translate into distinct biological outcomes, ranging from microbial antagonism to stronger attenuation of chitin-triggered host immunity”

Reviewer #3

The authors have adequately addressed the reviewers concerns.

Answer: We are happy that the current version of the manuscript meets your expectations. Thank you again for your helpful comments and feedback that helped to improve our study.

References

Z. Bozsoki, J. Cheng, F. Feng, K. Gysel, M. Vinther, K.R. Andersen, G. Oldroyd, M. Blaise, S. Radutoiu, & J. Stougaard, Receptor-mediated chitin perception in legume roots is functionally separable from Nod factor perception, *Proc. Natl. Acad. Sci. U.S.A.* 114 (38) E8118-E8127, <https://doi.org/10.1073/pnas.1706795114> (2017)

Tingting Liu *et al.* Chitin-Induced Dimerization Activates a Plant Immune Receptor. *Science* **336**,1160-1164(2012). DOI:[10.1126/science.1218867](https://doi.org/10.1126/science.1218867)

Zhang J, Sun J, Chiu CH, Landry D, Li K, Wen J, Mysore KS, Fort S, Lefebvre B, Oldroyd GED, Feng F. A receptor required for chitin perception facilitates arbuscular mycorrhizal associations and distinguishes root symbiosis from immunity. *Curr Biol.* 2024 Apr 22;34(8):1705-1717.e6. doi: 10.1016/j.cub.2024.03.015. Epub 2024 Apr 3. PMID: 38574729; PMCID: PMC11037463.